# Randomization Boosts KV Caching, Learning Balances Query Load: A Joint Perspective

**Fangzhou Wu**[1], **Sandeep Silwal**[1], **Qiuyi (Richard) Zhang**[2]
[1]University of Wisconsin–Madison, [2]Google DeepMind
`fwu89@wisc.edu, silwal@cs.wisc.edu, qiuyiz@google.com`

## Abstract

KV caching is a fundamental technique for accelerating Large Language Model (LLM) inference by reusing key-value (KV) pairs from previous queries, but its effectiveness under limited memory is highly sensitive to the eviction policy. The default Least Recently Used (LRU) eviction algorithm struggles with dynamic online query arrivals, especially in multi-LLM serving scenarios, where balancing query load across workers and maximizing cache hit rate of each worker are inherently conflicting objectives. We give the first unified mathematical model that captures the core trade-offs between KV cache eviction and query routing. Our analysis reveals the theoretical limitations of existing methods and leads to principled algorithms that integrate provably competitive randomized KV cache eviction with learning-based methods to adaptively route queries with evolving patterns, thus balancing query load and cache hit rate. Our theoretical results are validated by extensive experiments across 4 benchmarks and 3 prefix-sharing settings, demonstrating improvements of up to **6.92×** in cache hit rate, **11.96×** reduction in latency, **14.06×** reduction in time-to-first-token (TTFT), and **77.4%** increase in throughput over the state-of-the-art methods. Our code is available at `https://github.com/fzwark/KVRouting`.

## 1 Introduction

The increasing demands of Large Language Model (LLM) services impose substantial inference overhead on the LLM serving system (Jaillet et al., 2026). KV caching has emerged as a core technique to alleviate such costs by storing and reusing the key-value pairs of previously processed tokens as reusable prefixes for future generation (Vaswani et al., 2017). While practical, its effectiveness under limited memory is highly sensitive to the eviction policy, particularly in online processing, since the *cache hit rate* of a future query depends directly on which tokens have been evicted beforehand (Dan & Towsley, 1990). Although the traditional Least Recently Used (LRU) eviction policy (O'Neil et al., 1993; Fiat et al., 1991) remains the dominant choice in current LLM-serving system designs (Zheng et al., 2024; Kwon et al., 2023; NVIDIA, 2025), its strategy of evicting the oldest tokens within specific prefix-sharing cache structures is fragile and can be readily compromised under dynamic query arrivals (Fiat et al., 1991). In worst-case scenarios (Figure 1, left), LRU may evict exactly the tokens needed by the next query, leading to cache misses and ultimately increasing inference latency.

This instability of eviction becomes more pronounced in a scaled multi-LLM setting, where the objective of balancing query loads across LLMs inherently conflicts with the goal of maximizing the cache hit rate for the single model. Such conflicts naturally extend to a general ***KV cache-aware load balancing*** problem. Figure 1 (right) outlines its key trade-offs between *caching affinity* and *global load balancing*: (i) While maximizing caching affinity by routing similar queries to the same LLM may appear ideal, it inevitably leads to severe global load imbalance. This becomes particularly problematic when large batches of similar queries are routed to a single LLM, resulting in queueing delay dominating over actual service time and nullifying the benefits from the high cache hit rate. (ii) Conversely, distributing queries to balance query load without considering the cache situation leads to poor cache hit rate and, ultimately, suboptimal end-to-end inference latency. Furthermore, routing queries across LLMs introduces dependencies within the online sequence of queries, as actions taken in earlier steps have a sizeable impact on those taken later, creating additional variability, and further

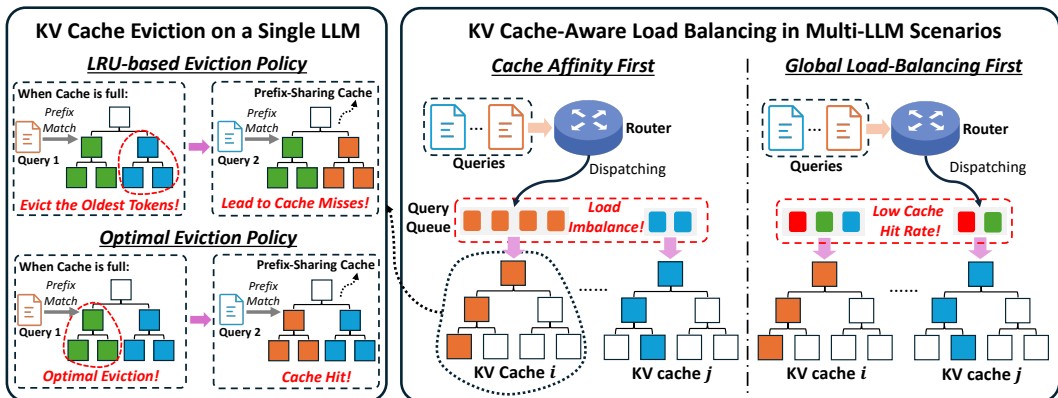

Figure 1: Key trade-offs between single LLM cache affinity and global load balancing in the KV cache-aware load balancing problem.

destabilizing existing LRU eviction policies. These complex tensions underscore the need for a formal mathematical model capturing the underlying dynamics.

Existing work, however, primarily relies on heuristics to address this problem, where they either adopt a static linear score that ranks each LLM based on cache hit rate and queue length, or employ a rule-based strategy that switches between highest-hit-rate and least-loaded routing using a predefined load-balance threshold (NVIDIA, 2025; llm d, 2025; Zheng et al., 2024). While easy to implement, these methods suffer from key limitations and struggle to optimize both objectives. They are inherently static and unable to adapt to dynamic query arrival patterns. Their modeling of queue workload is limited to the raw counts of pending queries, which is overly simplistic and fails to capture the true dynamics of system congestion. They are also fundamentally limited in achieving optimal performance, as they are rooted in heuristics, lack formal modeling, and therefore fail to capture the intricate coupling between cache eviction and load balancing.

This work establishes the first unified model of KV cache-aware load balancing to fill the gap between practical designs and theoretical understanding. The model formalizes the end-to-end latency of a query on each LLM by decomposing it into service time and queuing delay, explicitly modeling service time as a function of cache state and defining the queue load of each LLM as the cumulative service time of its assigned queries, with the objective of minimizing the makespan across LLMs. Within this formulation, our formal analysis reveals the inherent limitations of the LRU-based eviction policies, whose competitive ratio degrades to $O(n)$ in the worst case.

These insights further motivate the proposition of two principled algorithms, RLT and LBGR. For each LLM, RLT introduces *Randomized Leaf Token* eviction to replace traditional LRU, improving robustness to dynamic query arrival patterns and achieving a worst-case competitive ratio of $O(\log n)$. LBGR then routes queries greedily by estimating the end-to-end latency for each LLM using a dynamic, online learning-based strategy. In particular, it estimates the service time and queue load using a global cache tracker and applies exponential decay to the queue load estimate to account for its natural reduction over time. To capture residual latency beyond service and queueing delays, an online residual regression model is deployed and continuously updated during runtime.

We validate the effectiveness of our algorithms through extensive experiments across 4 benchmarks and 3 distinct prefix-sharing settings, demonstrating up to **11.96×** and **5.46×** reductions in end-to-end latency, and up to **14.06×** and **7.19×** reductions in time-to-first-token (TTFT) compared to state-of-the-art methods on Llama-3.1-8B-Instruct (NVIDIA L40) and Llama-3.1-70B-Instruct (NVIDIA H200), respectively, with context lengths up to 8k tokens. Furthermore, on Llama-3.1-8B-Instruct, RLT achieves up to a **6.92×** higher cache hit rate and **77.4%** higher throughput than the LRU-based eviction policy under worst-case query arrivals. To summarize, our main contributions are:

1. We present the first unified formal model for the KV cache-aware load balancing problem, which captures the intricate coupling between local cache eviction and global load balancing.

2. Within this model, we provide a formal analysis that reveals the theoretical limitations of existing system designs and motivates two simple yet effective algorithms, RLT and LBGR.

3. RLT introduces Randomized Leaf Token eviction in prefix-sharing KV cache structure, improving robustness to dynamic query arrivals and achieving a worst-case competitive ratio of $O(\log n)$, which is exponentially better than the $O(n)$ achieved by the existing LRU-based eviction policy.

4. LBGR employs an online regression model that incorporates cache state to dynamically estimate end-to-end latency and greedily route queries to the LLM with the lowest predicted latency, providing greater adaptability to dynamic query patterns than prior static heuristic-based methods.

5. We conduct extensive evaluations on 4 benchmarks across 3 prefix-sharing settings, which validate the effectiveness of the proposed algorithms, showing notable reductions in end-to-end latency.

## 2 BACKGROUND

**KV Cache Management.** Recent work on KV cache management falls into two strands. Context-aware methods leverage model-driven signals to compress the KV cache, retaining only a fixed-budget subset (Zhang et al., 2023; Xiao et al., 2024; Liu et al., 2023). Examples include H2O, which preserves high-attention tokens, and StreamingLLM, which retains initial attention sinks (Zhang et al., 2023; Xiao et al., 2024; Liu et al., 2023). Another strand aims to improve KV cache efficiency via memory layout and interface design (Kwon et al., 2023; Zheng et al., 2024). vLLM reduces fragmentation with fixed-size virtualized memory pages (Kwon et al., 2023), while RadixAttention enables prefix sharing and reuse via a radix tree (Zheng et al., 2024). However, those systems rely on LRU-based KV cache eviction, which is fragile under dynamic or adversarial query arrivals. The lack of formal analysis further reveals a gap between practical designs and theoretical understanding.

**KV Cache-Aware Load Balancing.** In multi-LLM serving, KV cache-aware load balancing arises from the need to balance queue load while preserving KV reuse (Sun et al., 2024; Zheng et al., 2024; Lee et al., 2024). Most systems use heuristics that trade off cache affinity and queue load (NVIDIA, 2025; Zheng et al., 2024; llm d, 2025). For instance, SGLang switches between hit-rate and load-based routing using a fixed threshold (Zheng et al., 2024), while others use static linear scores (NVIDIA, 2025; llm d, 2025). While practical, these methods are largely heuristic, lack formal modeling, and may underperform under dynamic query arrivals. See extended related work in Appendix D.

## 3 FORMAL PROBLEM FORMULATION AND THEORETICAL ANALYSIS

While modeling all aspects of the KV cache-aware load balancing is intractable, its essential properties can be effectively captured through a simplified formulation that minimizes the makespan across all LLMs during serving.

### 3.1 UNIFIED ONLINE FORMULATION FOR KV CACHE-AWARE LOAD BALANCING

Let $M$ denote the number of workers (LLMs) [1], indexed by $i \in [M]$, each denoted by $m_i$ and equipped with a KV cache of size $B_i$ tokens. We are given a sequence of $N$ queries, $Q = \{q_j\}_{j=1}^N$, which are processed in a fixed order (online arrival). Each query $q_j$ has an input length of $|q_j|$ tokens, and its response $a_j$ consists of $|a_j|$ tokens. We use two variables to track the state of each worker $m_i$ after processing $j$ queries: (1) $P_i^{(j)}$, which denotes the total accumulated load on worker $m_i$, and (2) $S_i^{(j)}$, which represents the cache state of worker $m_i$. The system is initialized at $j = 0$ with $P_i^{(0)} = 0$ and $S_i^{(0)} = \varnothing$ for all $i \in [M]$. For any query $q_j$, we define $h_{ij} = h(S_i^{(j-1)}, q_j)$ as the number of cache-hit tokens on $m_i$. Let $\alpha_{\text{CACHED}}$ and $\alpha_{\text{MISS}}$ denote the time costs of processing cached and uncached tokens, respectively. Then, the service time cost for $q_j$ on $m_i$ can be defined as

$$Cost_{ij} = \alpha_{\text{CACHED}} \cdot h_{ij} + \alpha_{\text{MISS}} \cdot (|q_j| - h_{ij}) + O_{ij} \qquad (1)$$

where $O_{ij}$ is the time cost of generating the output $a_j$. Let $x_{ij} \in \{0, 1\}$ be the binary decision variable indicating whether query $q_j$ is routed to worker $m_i$, subject to $\sum_{i=1}^M x_{ij} = 1, \forall j \in [N]$.

---

[1] Throughout, we use "LLM" and "worker" interchangeably and assume all deployed LLMs are of the same type.

Table 1: Notation and definitions.

| Notation | Description |
|---|---|
| $M$ | Number of workers (LLMs) |
| $m_i$ | The $i$-th worker (LLM) |
| $B_i$ | KV cache size (in tokens) of worker $m_i$ |
| $Q$ | Queries processed in a fixed order (online arrival) |
| $q_j$ | Input token sequence of $j$-th query |
| $a_j$ | Output token sequence of $j$-th query |
| $P_i^{(j)}$ | Query load of worker $m_i$ after processing $j$ queries |
| $S_i^{(j)}$ | Cache state of worker $m_i$ after processing $j$ queries |
| $h_{ij}$ | Number of cache-hit tokens of $q_j$ on worker $m_i$ |
| $\alpha_{\text{CACHED}}$ | Per-token time cost for cached tokens |
| $\alpha_{\text{MISS}}$ | Per-token time cost for uncached tokens |
| $Cost_{ij}$ | Service time cost of processing $q_j$ on worker $m_i$ |
| $x_{ij}$ | Binary routing variable, equal to 1 if $q_j$ is assigned to worker $m_i$ |

Once the assignment $x_{ij}$ is made for $q_j$, the system state is updated for all workers accordingly. The queue load of each worker is incremented by the service time cost only if it was assigned the query:

$$P_i^{(j)} = P_i^{(j-1)} + x_{ij} \cdot Cost_{ij}. \tag{2}$$

The cache state is updated only for the worker that processes the query:

$$S_i^{(j)} = \begin{cases} \text{UPDATECACHE}(S_i^{(j-1)}, q_j, B_i) & \text{if } x_{ij} = 1 \\ S_i^{(j-1)} & \text{if } x_{ij} = 0 \end{cases} \tag{3}$$

where $\text{UPDATECACHE}(\cdot)$ is a function (e.g., an LRU policy) that returns the new cache state of $m_i$.

The objective is to find the assignment $x = \{x_{ij}\}$ that minimizes the following *makespan*:
$\min_x \left( \max_{i \in [M]} \{P_i^{(N)}\} \right)$.

## 3.2 THEORETICAL LIMITATIONS OF LRU-BASED EVICTION

**Leaf-LRU.** Building on our unified model above, we formally analyze the performance of LRU-based eviction policies, focusing on Leaf-LRU (L-LRU), the eviction algorithm used in SGLang, one of the state-of-the-art LLM serving systems. In SGLang, the RadixAttention mechanism stores KV values in a radix tree structure organized at token granularity, and evicts the least recently used leaf tokens using L-LRU when memory is saturated. Since the process time of token hits is negligible compared to that of misses, we ignore the hit cost and focus on the total number of token misses.

**Cache Matching and Arrival Model.** As the response $a_j$ of each query $q_j$ is also cached in the radix tree, we define the *complete token path* $\Gamma_j := q_j \| a_j$ as the concatenation of the prompt and its generated tokens. Accordingly, we extend the query set $Q$ to the set of complete token paths $\widetilde{Q} := \{\Gamma_j\}_{j=1}^N$. Cache matching then reduces to longest-prefix path matching in the radix tree. In the following analysis, we assume an arbitrary (potentially adversarial) arrival order over $\widetilde{Q}$.

**Competitive Ratio of Leaf-LRU in RadixAttention.** We compare L-LRU against the optimal eviction strategy (OPT), which evicts the leaf token whose next appear lies furthest in the future. Our analysis covers both single-query and batch processing settings. Following (Fiat et al., 1991), we partition $\widetilde{Q}$ into disjoint phases $\{\P_v\}$, where each phase $\P_v$ (except possibly the last) contains exactly $B_i$ distinct tokens. In any phase $\P_v$ ($v \geq 2$), a token is defined as *clean* for L-LRU if it was not in the cache at the end of $\P_{v-1}$ and has not yet appeared in $\P_v$; *new* if it appears for the first time in $\P_v$; and *old* if it has already been seen earlier in $\P_v$. Inspired by (Fiat et al., 1991), we begin by establishing a lower bound on the number of misses incurred by OPT during a phase in the following Lemma 1 [2].

**Lemma 1.** *In RadixAttention, the amortized number of misses incurred by OPT in a phase $\P_v$ with $c$ clean tokens is lower bounded by $\max\{c/2, 1\}$.*

---

[2]All proofs are provided in Appendix C.

We then analyze the maximum number of misses that L-LRU can incur in a phase $\P_v$ with $c$ clean tokens. Misses can occur in only two cases: (1) when a new token appears, or (2) when a previously evicted old token reappears. We first derive an upper bound on the number of misses assuming that case (2) does not occur (Lemma 2), and then show that case (2) indeed never arises (Lemma 3).

**Lemma 2.** *For any phase $\P_v$ with $c$ clean tokens, the number of misses incurred by L-LRU under single-query processing is at most $B_i - \mathcal{L} + c$, assuming no old token reappears after eviction.*

**Lemma 3.** *Under single-query processing in L-LRU, no old token reappears in any $\P_v$ after eviction.*

Using Lemmas 1 to 3, we establish the competitive ratio of L-LRU under single-query processing.

**Theorem 4** (Single-Query). *Under single-query processing setting, the competitive ratio of L-LRU in RadixAttention on worker $m_i$ with cache capacity $B_i$ is upper bounded by $(B_i - \mathcal{L} + 2)$ and lower bounded by $(B_i - \mathcal{L} + 1)$, where $\mathcal{L}$ denotes the minimum length over all $\Gamma_j \in \widetilde{Q}$.*

We next consider a continuous batching setting, where the system handles $\beta$ distinct queries concurrently during processing. We assume $\beta \mathcal{L}_{max} \leq B_i$ to ensure that the cache can store the KV entries of all $\beta$ in-progress queries, each with length up to $\mathcal{L}_{max}$, without evicting any entries belonging to those queries. This capacity condition aligns with typical real-world configurations and is precisely the regime where the choice of eviction policy has a significant impact. Furthermore, we assume the queries within each batch are distinct, which is essential for obtaining a meaningful worst-case analysis in the batched setting (see Appendix C for a detailed discussion). Building on the previous analysis, we present the competitive ratio of L-LRU under this setting in the following Theorem 5.

**Theorem 5** (Batch). *Consider the continuous batch setting with batch size $\beta$. Let $\mathcal{L}_{max}$ and $\mathcal{L}$ denote the maximum and minimum lengths over all $\Gamma_j \in \widetilde{Q}$. If $\beta \mathcal{L}_{max} \leq B_i$, where $B_i$ is the cache capacity of worker $m_i$, and all queries in a batch are distinct, then, the competitive ratio of L-LRU in RadixAttention is upper bounded by $(B_i - \mathcal{L} - \beta + 3)$ and lower bounded by $(B_i - \mathcal{L} - \beta + 2)$.*

Our analysis (Theorems 4 and 5) shows that L-LRU achieves an $O(n)$ competitive ratio in the worst case. When fixing the KV cache size $B_i$ and the maximum length $\mathcal{L}_{max}$, decreasing the minimum length $\mathcal{L}$ drives the competitive ratio toward $B_i$. This indicates that when lengths of queries are highly imbalanced, the competitive ratio increases and the performance of L-LRU may degrade.

## 4 METHODOLOGY

Motivated by the formal model and analysis above, we integrate a randomized eviction algorithm (Section 4.1) with a learning-based greedy routing algorithm that routes queries based on estimated end-to-end latency (Section 4.2) to balance query load while maintaining a high cache hit rate.

| **Algorithm 1** RLT | **Algorithm 2** LBGR |
|---|---|
| 1: Marking token set initialization: $\mathcal{T} \leftarrow \varnothing$ | 1: ▷ /* Online Routing Thread */ |
| 2: KV cache size: $B_i$ | 2: **for** $q_j \in Q$ **do** |
| 3: Complete token paths: $\widetilde{Q} \leftarrow Q$ | 3:    **for** $i \in [M]$ **do** |
| 4: **for** $\Gamma_j \in \widetilde{Q}$ **do** | 4:       Estimate the $\widetilde{h}_{ij}$ via global radix tree |
| 5:    $S_i^{(j)} \leftarrow S_i^{(j-1)}$ | 5:       $\widehat{Cost}_{ij} \leftarrow \alpha_{\text{CACHED}} \widetilde{h}_{ij} + \alpha_{\text{MISS}}(|q_j| - \widetilde{h}_{ij})$ |
| 6:    **for** $t \in \Gamma_j$ **do** | 6:       $\phi_{ij} \leftarrow \phi(\widetilde{h}_{ij}, |q_j| - \widetilde{h}_{ij}, \widetilde{P}_i^{(j-1)})$ |
| 7:       $\mathcal{T} \leftarrow \mathcal{T} \cup \{t\}$ | 7:       $\widehat{E}_{ij} \leftarrow \widehat{Cost}_{ij} + \widetilde{P}_i^{(j-1)} + \theta_i^\top \phi_{ij}$ |
| 8:       **if** $|\mathcal{T}| = B_i + 1$ **then** | 8:    $i^* \leftarrow \arg\min_{i \in [M]} \widehat{E}_{ij}, x_{ij} := \mathbf{1}[i = i^*]$ |
| 9:          $\mathcal{T} \leftarrow \{t\}$ | 9:    $\forall i \in [M], \widetilde{P}_i^{(j)} \leftarrow \widetilde{P}_i^{(j-1)} + x_{ij} \widehat{Cost}_{ij}$ |
| 10:       **if** $t \in S_i^{(j)}$ **then** | 10:    Route $q_j$ to $m_{i^*}$ |
| 11:          **continue** | 11: ▷ /* Online Updating Thread */ |
| 12:       **else** | 12: **if** observe actual latency $E_{ij}$ **then** |
| 13:          **if** $S_i^{(j)}$ is full **then** | 13:    $\theta_i \leftarrow \text{ONLINEUPDATE}(\theta_i, \phi_{ij}, E_{ij} - \widehat{E}_{ij})$ |
| 14:             $U \leftarrow \{\text{leaf tokens in } S_i^{(j)}\} \setminus \mathcal{T}$ | 14:    $\widetilde{P}_i \leftarrow \text{RELEASELOAD}(\widetilde{P}_i, \widehat{Cost}_{ij}, \rho, \Delta t)$ |
| 15:             Choose $u \in U$ uniformly at random | 15: ▷ /* Background Decay Thread */ |
| 16:             $S_i^{(j)} \leftarrow \text{EVICT}(S_i^{(j)}, u)$ | 16: Every $\Delta t$ time units: $\forall i, \widetilde{P}_i \leftarrow \rho \widetilde{P}_i$ |
| 17:       $S_i^{(j)} \leftarrow \text{LOAD}(S_i^{(j)}, t)$ | |

## 4.1 RANDOMIZED LEAF TOKEN EVICTION

Inspired by the classical marking algorithm (Fiat et al., 1991), we introduce the **Randomized Leaf Token** eviction algorithm, RLT. As detailed in Algorithm 1, RLT iterates over each token in every path $\Gamma_j \in \widetilde{Q}$ and marks accessed tokens by adding them to a marking set $\mathcal{T}$ (lines 4–7 in Algorithm 1). Once $B_i + 1$ unique tokens have been marked, all marks are cleared except for the most recently accessed token (lines 8–9 in Algorithm 1). When a requested token is not in the cache and the cache is full, RLT evicts an unmarked leaf token uniformly at random from the current cache before loading the new token (lines 13-17 in Algorithm 1).

**Competitive Ratio of RLT.** The effectiveness of RLT stems from its use of randomness, which breaks the dependence on query arrival order and enhances robustness against dynamic or adversarial query arrivals, yielding more stable performance. We formally establish its competitive ratio in both single-query and batch processing settings in the following Theorem 6 and corollary 7.

**Theorem 6** (Single-Query). *RLT is $\Theta(\log(B_i - \mathcal{L}))$-competitive on worker $m_i$ with cache capacity $B_i$ under single-query processing setting, where $\mathcal{L}$ is the minimal length over all $\Gamma_j \in \widetilde{Q}$.*

**Corollary 7** (Batch). *RLT is $\Theta(\log(B_i - \mathcal{L} - \beta))$-competitive on worker $m_i$ with capacity $B_i$ under continuous batching setting, where $\mathcal{L}$ is the minimal length over all $\Gamma_j \in \widetilde{Q}$, and $\beta$ is the batchsize.*

Our analysis shows that RLT achieves a competitive ratio of $O(\log n)$, which is a logarithmic improvement over L-LRU. We further prove that no dependent algorithm can achieve a better competitive ratio than RLT in either single-query or continuous batching settings (Theorem 8 and corollary 9).

**Theorem 8** (Single-Query). *No randomized eviction algorithm can achieve a competitive ratio better than $\Theta(\log(B_i - \mathcal{L}))$ on worker $m_i$ with cache capability of $B_i$ in the single-query processing setting, where $\mathcal{L}$ denotes the minimal length over all $\Gamma_j \in \widetilde{Q}$.*

**Corollary 9** (Batch). *No randomized eviction algorithm can achieve a competitive ratio better than $\Theta(\log(B_i - \mathcal{L} - \beta))$ on worker $m_i$ with cache capability of $B_i$ in the continuous batching setting, where $\mathcal{L}$ denotes the minimal length over all $\Gamma_j \in \widetilde{Q}$ and $\beta$ is the batch size.*

## 4.2 LEARNING-BASED GREEDY ROUTING

To handle dynamically evolving arrivals, we introduce the **Learning-Based Greedy Routing** (LBGR) algorithm (Algorithm 2), which estimates per-LLM end-to-end latency via an online regression and greedily routes each query to the LLM with the lowest estimate.

**Overall Cost Formulation.** The end-to-end latency $E_{ij}$ of routing query $q_j$ to worker $m_i$ can be modeled in the decomposition of $E_{ij} = Cost_{ij} + P_i^{(j-1)}$. Building on this, LBGR estimates this latency via the following predictive model:

$$\widehat{E}_{ij} = \underbrace{\widehat{Cost}_{ij}}_{\text{service time estimation}} + \underbrace{\widetilde{P}_i^{(j-1)}}_{\text{queue load estimation}} + \underbrace{\theta_i^\top \phi_{ij}}_{\text{residual correction}} \tag{4}$$

where $\widehat{Cost}_{ij}$ is the estimated service time, $\widetilde{P}_i^{(j-1)}$ is the current estimated queue load of $m_i$ (see below), and $\theta_i^\top \phi_{ij}$ is a learned regression term (see below) that captures residual latency bias.

**Service Time Estimation.** Building on the global radix tree in SGLang (which records the cache state of each worker and enables efficient estimation of cache hit rates; see Appendix E for futher discussion), we estimate the number of cache-hit tokens for query $q_j$ on worker $m_i$ as $\widetilde{h}_{ij}$, serving as a proxy for the true hit count $h_{ij}(S_i^{(j-1)}, q_j)$ under cache state $S_i^{(j-1)}$. The estimated service time is then:

$$\widehat{Cost}_{ij} = \alpha_{\text{CACHED}} \cdot \widetilde{h}_{ij} + \alpha_{\text{MISS}} \cdot \left(|q_j| - \widetilde{h}_{ij}\right) \tag{5}$$

where we omit the time cost of output token generation and absorb it into the learned residual term.

**Queue Load Estimation.** To approximate queue load, each $m_i$ maintains a queue load $\widetilde{P}_i$ that is updated with each incoming query. When $q_j$ is routed to $m_i$ ($x_{ij} = 1$), the load is incremented as:

$$\widetilde{P}_i^{(j)} = \widetilde{P}_i^{(j-1)} + x_{ij} \cdot \widehat{Cost}_{ij}. \tag{6}$$

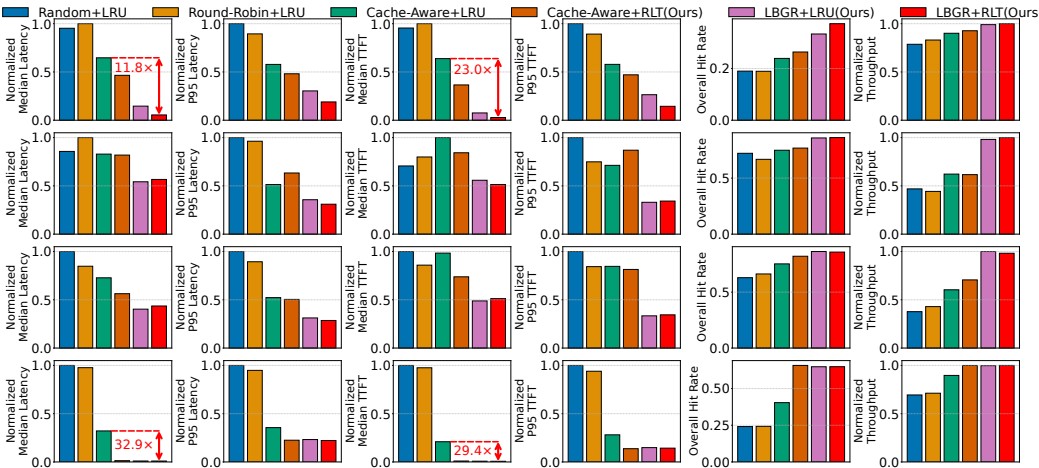

Figure 2: Results on Llama-3.1-8B-Instruct (Latency, TTFT, and Throughput normalized for comparison). Rows correspond to: GSP (top), ShareGPT (second), UltraChat (third), and Loogle (bottom). For the first four metrics, lower is better; for the last two, higher is better. Our algorithms consistently outperform all baselines across all benchmarks and metrics.

To further simulate the natural reduction in queue load over time, we apply exponential decay independently every $\Delta t > 0$ time units: $\widetilde{P}_i \leftarrow \rho \widetilde{P}_i$, where $\rho \in (0, 1)$ is the decay factor. Upon completion of $q_j$, LBGR releases any remaining load associated with $q_j$ (line 13 in Algorithm 2).

**Greedy Routing with Online Residual Correction.** Latency bias naturally arises from complex environmental factors that are not captured by service time or queue load. To account for these fluctuations, we learn a lightweight residual model for each $m_i$ using online linear regression parameterized by $\theta_i$. For each query $q_j$, we extract the feature vector $\phi_{ij} := \phi\big(\widetilde{h}_{ij}, |q_j| - \widetilde{h}_{ij}, \widetilde{P}_i^{(j-1)}\big)$, and predict end-to-end latency $\widehat{E}_{ij}$ for all workers $i \in [M]$ using Equation (4). The query $q_j$ is then dispatched to the worker with the lowest estimated latency, $i^* = \arg\min_{i \in [M]} \widehat{E}_{ij}$. Upon observing the realized latency $E_{ij}$, the model $\theta_i$ is updated online by minimizing the squared loss $\big(E_{ij} - \widehat{E}_{ij}\big)^2$.

## 5 EVALUATION

**Models.** Following (Zheng et al., 2024), we evaluate two types of LLMs: dense Llama-3.1 models (Grattafiori et al., 2024) and the sparse (MoE) Mixtral model (Jiang et al., 2024), with model sizes ranging from 8B to 70B. We vary the number of deployed workers from 1 to 10, with 4 workers used as the default setting. All experiments for Llama-3.1-8B-Instruct are run on 10 NVIDIA L40 GPUs, while experiments with Llama-3.1-70B-Instruct and Mixtral 8×7B use 4 NVIDIA H200 GPUs.

**Baselines.** For the eviction policy, we compare RLT with the default L-LRU used in SGLang (state-of-the-art LLM serving system). For load balancing, we evaluate LBGR against three routing algorithms: (i) random routing, (ii) round-robin routing (SGLang, 2025c), and (iii) cache-aware routing (SGLang, 2025a). These combinations yield three baselines: (1) Random+LRU [3], (2) Round-Robin+LRU, and (3) Cache-Aware+LRU, where Cache-Aware+LRU is **the current state-of-the-art**.

**Workloads.** Following (Zheng et al., 2024), we evaluate over 3 distinct prefix-sharing workloads under limited cache memory, spanning both *synthetic* and *real-world* scenarios: (1) Synthetic prefix-caching test using the Generated Shared Prefix (GSP) benchmark (SGLang, 2025b); (2) Multi-turn conversations using real-world logs from ShareGPT (sha, 2023) and UltraChat (Ding et al., 2023); (3) Long-document QA using Loogle (Li et al., 2024). We extend these benchmarks by introducing variability in prompt lengths to simulate realistic and challenging serving conditions. The number of output tokens is varied from 4 to 128, with 4 used as the default. Furthermore, we consider two distinct query arrival orders: (i) a random query order, and (ii) a worst-case round-robin order, with

---

[3] We use LRU to denote L-LRU in baseline names for simplicity.

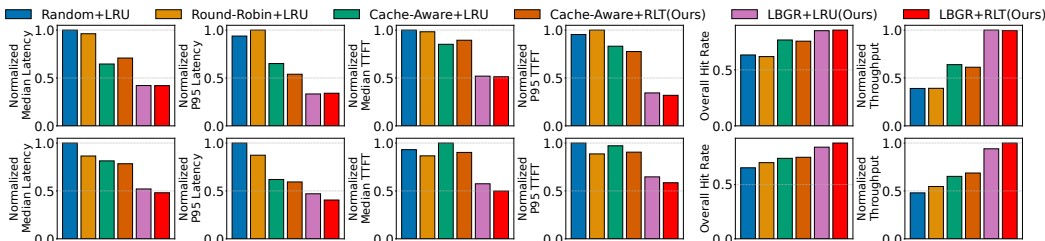

Figure 3: Results on Llama-3.1-70B-Instruct (top) and Mixtral-8×7B-Instruct-v0.1 (bottom) under the ShareGPT benchmark. Our algorithms consistently outperform all baselines across metrics.

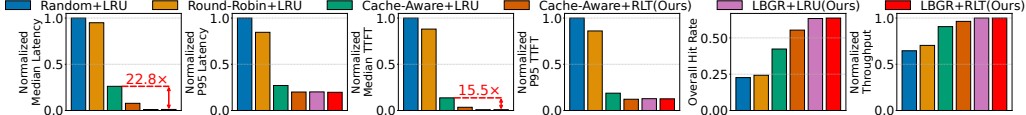

Figure 4: Results on Llama-3.1-8B-Instruct under the Loogle benchmark with the worst-case round-robin arrival order. Round-robin alternates queries to disrupt KV locality. LBGR+RLT counters this, outperforming all baselines across all metrics.

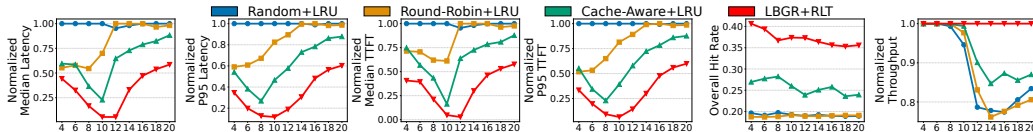

Figure 5: Results on Llama-3.1-8B-Instruct under GSP benchmark with varying request rates.

the random order used by default. All workloads follow a Poisson arrival process, and we vary the request rate from 4 to 20 requests/s, with 12 requests/s used as the default.

**Metrics.** We report four main performance metrics: cache hit rate, throughput, latency, and time to first token (TTFT). For latency and TTFT, we report the median (P50) to reflect the typical user experience and the 95th percentile (P95) to capture tail latency. Furthermore, we provide a fine-grained breakdown of runtime overhead, reporting time cost for eviction and routing operations. For additional experimental setup details, see Appendix A.

**End-to-End Performance.** We present the main results on Llama-3.1-8B-Instruct across 4 benchmarks in Figure 2. Our methods, Cache-Aware+RLT, LBGR+LRU, and LBGR+RLT, consistently outperform all baselines, **achieving the lowest latency and TTFT on every benchmark**. On average, LBGR+RLT achieves **30.9×** and **44.49×** improvements in median latency and TTFT compared to Random+LRU, and improves over the state-of-the-art Cache-Aware+LRU by **11.96×** in median latency and **14.06×** in median TTFT. For tail performance, it still achieves an average **2.03×** improvement in P95 latency and **2.62×** speedup in P95 TTFT over Cache-Aware+LRU. When applying RLT, Cache-Aware+RLT achieves **6.98×** and **6.40×** faster than Cache-Aware+LRU in median latency and TTFT on average. With the learning-based strategy, LBGR+LRU also outperforms Cache-Aware+LRU, achieving **9.98×** lower median latency and **9.58×** lower median TTFT on average. Furthermore, LBGR+RLT achieves the highest cache hit rate and throughput across all benchmarks, averaging **36.45%** higher hit rate, **36.51%** higher throughput than Cache-Aware+LRU. These results demonstrate the strong and comprehensive efficiency gains of our algorithms under dynamic query arrivals with varying lengths.

**Model Size & Architecture.** We evaluate the generalizability of our algorithms across model scales and architectures, using Llama-3.1-70B-Instruct as a representative large dense model and Mixtral-8×7B-Instruct-v0.1 as a sparse MoE model. As shown in Figure 3, our methods consistently outperform all baselines on the ShareGPT benchmark across all metrics. LBGR+RLT achieves the lowest latency and TTFT, reducing median latency by **38%** and TTFT by **44.87%** compared with the best baseline, and also achieves the highest hit rate and throughput on both models. These results demonstrate the generalizability and adaptability of our algorithms serving both large dense and sparse MoE models. Additional results on other benchmarks are provided in Appendix B.

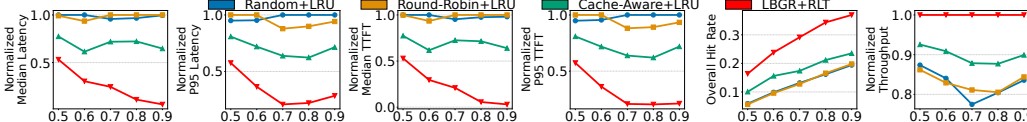

Figure 6: Results on Llama-3.1-8B-Instruct under GSP benchmark with varying number of workers (LLMs), where the $x$-axis denotes the number of workers.

Figure 7: Results on Llama-3.1-8B-Instruct under GSP benchmark with varying size of KV cache, where the $x$-axis denotes the percentage of GPU memory used by the KV cache.

Table 2: Ablation comparison of performance and runtime overhead between Cache-Aware+LRU and our methods on the GSP benchmark. Time-based metrics ($\downarrow$) are reported in milliseconds (ms), while hit rate ($\uparrow$) and throughput ($\uparrow$) are measured in percentage and requests per second, respectively.

| Method | P50 Latency | P95 Latency | P50 TTFT | P95 TTFT | Hit Rate | Throughput | Average Eviction Time | Average Routing Time |
|---|---|---|---|---|---|---|---|---|
| Cache-Aware+LRU | 26680.55 | 46766.77 | 25022.76 | 46139.36 | 23.89% | 10.73 | 0.13 | **0.47** |
| **Cache-Aware+RLT (Ours)** | 19191.25 | 38917.27 | 14332.81 | 37504.69 | 26.36% | 11.05 | 0.71 | 0.51 |
| **LBGR+LRU (Ours)** | 6025.11 | 24561.47 | 2958.01 | 21073.78 | 33.33% | 11.80 | **0.09** | 1.03 |
| **LBGR+RLT (Ours)** | **2263.61** | **15334.89** | **1088.57** | **11495.05** | **37.31%** | **11.92** | 1.05 | 1.45 |

**Query Order.** We evaluate the robustness of our algorithms on Loogle under worst-case query arrivals, where queries from each document arrive in a round-robin manner. As shown in Figure 4, LBGR+RLT maintains the **strongest** performance in the worst-case setting, achieving the **lowest** median and P95 latency and TTFT, and the **highest** hit rate and throughput compared with all baselines. Notably, it reduces median latency and TTFT by **22.8×** and **15.5×** compared to Cache-Aware+LRU. These results highlight the robustness of our algorithms under adversarial query arrivals.

**Request Rate.** We vary the request rate from 4 to 20 to evaluate the robustness of our method under different query arrival intensities. As shown in Figure 5, LBGR+RLT consistently achieves the **lowest** latency and TTFT at both median and P95 across all request rates. It also consistently delivers the **highest** cache hit rate and throughput across all settings. These results underscore the strong performance and robustness of our method under varying query loads.

**Number of Workers.** We assess the performance of our algorithms under different numbers of deployed workers (LLMs), varying the number of workers from 2 to 10. As shown in Figure 6, LBGR+RLT consistently outperforms all baselines across all metrics and settings. Even with as few as 2 workers, it maintains leading performance. For throughput, our method is higher when the number of workers is relatively small (e.g., 2 or 4). As the number of workers increases ($\geq 6$), the throughput of all methods converges to the same value. This is expected: with the request rate fixed at 12 requests/s, once the system has sufficient capacity, all methods can saturate this limit. These results underscore the scalability of our algorithms across varying numbers of deployed workers.

**KV Cache Size.** We evaluate the impact of the KV cache size by varying the allocated GPU memory percentage from 50% to 90% on L40. As shown in Figure 7, our algorithms consistently outperform all baselines across all metrics and cache size settings. As the cache size increases, the performance gap between our method and the baselines widens significantly. Even under constrained settings (e.g., using only 50% of GPU memory), our method maintains a clear lead. These results demonstrate the robustness and strong performance of our algorithms under varying KV cache availability.

**Ablation Study on the Effectiveness and Overhead of RLT and LBGR.** Table 2 presents a detailed analysis of the effectiveness and runtime overhead of our algorithms. Using RLT, Cache-Aware+RLT consistently outperforms Cache-Aware+LRU across all metrics, reducing median latency and TTFT by **28.1%** and **42.7%**, respectively. This confirms the effectiveness of RLT under dynamic and length-imbalanced query arrivals. Furthermore, adopting the learning-based routing strategy, LBGR+LRU

achieves even greater gains over Cache-Aware+LRU, reducing median latency and TTFT by **77.4%** and **88.2%**, respectively. This highlights the adaptiveness and effectiveness of LBGR compared to static routing strategies used in SGLang. Combining both algorithms, LBGR+RLT achieves the **best** performance, reducing median latency and TTFT by **11.8×** and **23×**, respectively. This shows that integrating learning-based routing with randomized eviction provides strong robustness and performance. In terms of runtime overhead, the total added runtime overhead of RLT and LBGR is only about **2ms** per query, which is negligible compared to overall end-to-end latency. Specifically, for eviction, RLT adds only 0.58ms over LRU with Cache-Aware, and 0.96ms when paired with LBGR. For routing, LBGR adds only 0.56ms over Cache-Aware when used with LRU, and 0.94ms when used with RLT. This demonstrates the practicality of our algorithms for real-world deployment.

**More Experiments.** We present additional results by varying the shared-prefix ratio, number of serving queries, output token length, and maximum batch size, where our methods achieve the best performance across all settings (Appendix Figures 10 to 13). We also conduct two ablations: (i) comparing RLT and L-LRU on a single worker, where RLT shows clear improvements (Appendix Table 3), and (ii) evaluating the impact of decay interval $\Delta t$, which highlights the importance of timely queue load decay for effective query routing (Appendix Figure 14). See Appendix B for details.

## 6    LIMITATIONS AND CONCLUSION

This work presents the first unified mathematical model that captures the core tensions between KV cache eviction and query routing in the KV cache-aware load balancing problem. Our analysis identifies the theoretical limitations of existing approaches and leads to principled algorithms that combine provably competitive randomized eviction with learning-based methods for adaptive query routing under dynamic workloads. We validate proposed algorithms through extensive experiments across 4 benchmarks and 3 different prefix-sharing settings, demonstrating substantial improvements in inference efficiency and notable reductions in end-to-end latency.

One limitation of our current implementation is the lack of evaluation for multi-modal inference. We focus on text-only KV cache and implement our algorithms based on the SGLang codebase, leaving support for multi-modality for future work. Furthermore, our available computation resources limit the experiments to at most 10 workers in a single-domain setup, which may not fully capture the behavior in larger or geo-distributed deployments. Exploring broader scales and additional domains is left for future investigation.

## REPRODUCIBILITY STATEMENT

We have taken concrete steps to ensure reproducibility of our results. An anonymized repository is included in the supplemental materials, containing the complete source code, configuration files, and scripts necessary to reproduce the experiments. All implementation details, such as model versions, hardware specifications, and evaluation procedures, are described in Section 5 and Appendix A. The design and implementation of our proposed algorithms are presented in Section 4 and Appendix A. For theoretical results, complete and detailed proofs are provided in Appendix C.

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

# A EXPERIMENTAL DETAILS

**Models.** Following (Zheng et al., 2024), we evaluate two types of LLMs: dense Llama-3.1 models (Grattafiori et al., 2024) and the sparse (MoE) Mixtral model (Jiang et al., 2024), with model sizes ranging from 8B to 70B. The number of deployed workers varies between 1 and 10 during evaluation, and is set to 4 by default unless otherwise specified. All experiments for Llama-3.1-8B-Instruct are run on 10 NVIDIA L40 GPUs, while experiments with Llama-3.1-70B-Instruct and Mixtral $8\times7$B use 4 NVIDIA H200 GPUs. Unless otherwise specified, we fix the KV cache size to approximately 200k tokens for each model to ensure a fair comparison. We use BF16 precision for all models, and apply quantization to FP8 for Llama-3.1-70B to accommodate memory limitations and maintain a 200k-token KV cache. With these settings, the 200k-token KV cache fits naturally within the maximum available GPU memory for Llama-3.1-Instruct-8B on L40 and Llama-3.1-Instruct-70B on H200 without requiring manual adjustment. For Mixtral, we manually configure the KV memory budget to use 80% of the GPU memory to support a comparable cache size. Unless otherwise specified, we allow each worker to automatically maximize the running batch size (maximum running queries) subject to the available GPU memory.

**Baselines.** For the eviction policy, we compare our randomized eviction algorithm, RLT, against the default adopted L-LRU eviction strategy in SGLang. This choice is motivated by two main reasons: (i) LRU-based eviction is the dominant approach in existing systems (Zheng et al., 2024; Kwon et al., 2023; llm d, 2025; NVIDIA, 2025) and thus represents the mainstream design choice, and (ii) Our implementation builds on the SGLang codebase, which is the state-of-the-art open-source LLM serving system, and uses L-LRU as its built-in eviction policy. Therefore, comparing against L-LRU is both reasonable and representative, and also demonstrates the effectiveness of our proposed method. For load balancing, we evaluate LBGR against three routing algorithms: (i) random routing, (ii) round-robin routing (SGLang, 2025c), and (iv) cache-aware routing (SGLang, 2025a). Round-robin routing cycles through workers in order, whereas cache-aware routing switches between the highest-hit-rate and the least-loaded routing based on a predefined heuristic load-balance threshold. These combinations yield three baselines: (1) Random+LRU, (2) Round-Robin+LRU, and (3) Cache-Aware+LRU, where Cache-Aware+LRU is the current state-of-the-art.

**Workloads.** Following prior work (Zheng et al., 2024), we evaluate over 3 distinct prefix-sharing workloads under limited cache memory, spanning both *synthetic* and *real-world* scenarios: (1) Synthetic prefix-caching test using the Generated Shared Prefix (GSP) benchmark (SGLang, 2025b); (2) Multi-turn conversations using real-world logs from ShareGPT (sha, 2023) and UltraChat (Ding et al., 2023); (3) Long-document QA using Loogle (Li et al., 2024). We extend these benchmarks by introducing variability and imbalance in prompt lengths to simulate realistic and challenging serving conditions. The number of output tokens is varied from 4 to 128, with 4 used as the default.

For the GSP benchmark, we consider 128 groups, each containing 32 queries (a total of 4096 queries) that share the same prefix and prompt length, differing only in their suffixes. To introduce prompt-length imbalance, we assign lengths cyclically across groups using 5 values spanning 3 representative scales: small (512 tokens), medium (1024 and 2048 tokens), and large (4096 and 8192 tokens). All groups share the same prefix ratio, with 4 settings evaluated (0.3, 0.5, 0.7, 0.9) and 0.5 used as the default. For the multi-turn conversation setting, we evaluate two benchmarks, ShareGPT and UltraChat, using 128 clients, each maintaining an independent multi-turn conversation. To capture the imbalance and dynamic nature of real-world conversations, we vary the number of conversation rounds per client across {2, 4, 6, 8}, with each round introducing a 1024-token user input (padded as needed to reach the target length). In the long-document QA task on the Loogle benchmark, we randomly select 512 documents, each paired with the full set of questions. To reflect input-length imbalance, we vary the document length by truncating each one to one of four target lengths: {1024, 2048, 4096, 8192} tokens.

Furthermore, we consider two distinct query arrival orders: (i) a random query order, and (ii) a worst-case round-robin order. Unless otherwise specified, we use the random order as the default setting. For the GSP and Loogle benchmarks, we additionally evaluate under a round-robin order, where we iterate over groups (or documents) in a fixed cycle and issue one query per group/document each turn, repeating this cycle until all queries have been dispatched. All workloads follow a Poisson arrival process, and we vary the request rate from 4 to 20 requests/s, with 12 requests/s used as the default.

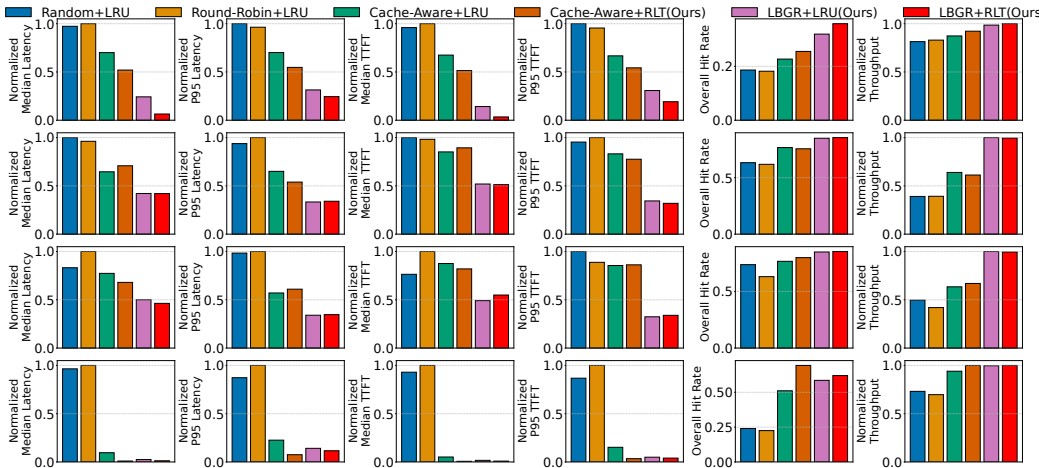

Figure 8: Results on Llama-3.1-70B-Instruct (Latency, TTFT, and Throughput normalized for comparison). Rows correspond to: GSP (top), ShareGPT (second), UltraChat (third), and Loogle (bottom). For the first four metrics, lower is better; for the last two, higher is better. Our algorithms consistently outperform all baselines across all benchmarks and metrics.

**Metrics.** We report four main performance metrics: cache hit rate, throughput, latency, and time to first token (TTFT). For latency and TTFT, we report median (P50) and P95. Furthermore, we provide a fine-grained breakdown of runtime overhead, including the time cost for eviction operations in L-LRU and RLT, and the routing operations of LBGR.

**Implementation.** Our implementation is based on the SGLang-0.4.6 codebase, a state-of-the-art open-source LLM serving system. We implement RLT in Cython to minimize eviction overhead, and integrate LBGR into SGLang's existing Rust-based cache-aware routing framework as a drop-in replacement policy. In the implementation of LBGR, we scale the overall time cost estimation to a per-1k-token unit to improve regression stability and reduce sensitivity to noise. For service time estimation ($\widehat{Cost_{ij}}$), we fix the cached token coefficient to $\alpha_{\text{CACHED}} = 0$ ms per 1k tokens and the uncached token coefficient to $\alpha_{\text{MISS}} = 1000$ ms per 1k tokens. For the background decay thread, we smooth the worker queue load using exponential decay with factor $\rho = 31/32$, updated every 20ms. For the online residual model, we use a 4-dimensional linear regression model with three input features and one bias term, updated online with a learning rate of 0.992.

# B  ADDITIONAL EXPERIMENTAL RESULTS

**Model Size & Architecture.** We evaluate the generalizability of our algorithms across model scales and architectures, using Llama-3.1-70B-Instruct as a representative large dense model and Mixtral-8×B-Instruct-v0.1 as a sparse MoE model. As shown in Figure 8 and Figure 9, our methods consistently outperform all baselines across all benchmarks and evaluation metrics. LBGR+RLT achieves the lowest average latency and TTFT, reducing median latency by **5.46×** and TTFT by **7.19×** compared to the best baseline on Llama-3.1-70B-Instruct, and reducing both median latency and TTFT by **24.2%** on Mixtral. It also achieves the highest average cache hit rate and throughput on both models. These results demonstrate the robustness and adaptability of our algorithms in serving both large dense and sparse MoE models.

**Shared-Prefix Ratio & Number of Shared-Prefix Queries.** We evaluate the impact of two key factors related to prefix sharing on algorithm performance: the shared prefix ratio (i.e., the fraction of tokens shared across queries in a group, see Appendix A for details), and the number of shared-prefix queries per group. As shown in Figure 10, when varying the shared prefix ratio from 0.3 to 0.9 on the GSP benchmark, our method consistently achieves strong performance across all settings. Even at low sharing (ratio = 0.3), our method outperforms all baselines by a clear margin, and at high sharing (ratio = 0.9), it continues to maintain the advantage. We further vary the number of queries per shared-prefix group from 4 to 128. Figure 11 shows that LBGR+RLT consistently outperforms all

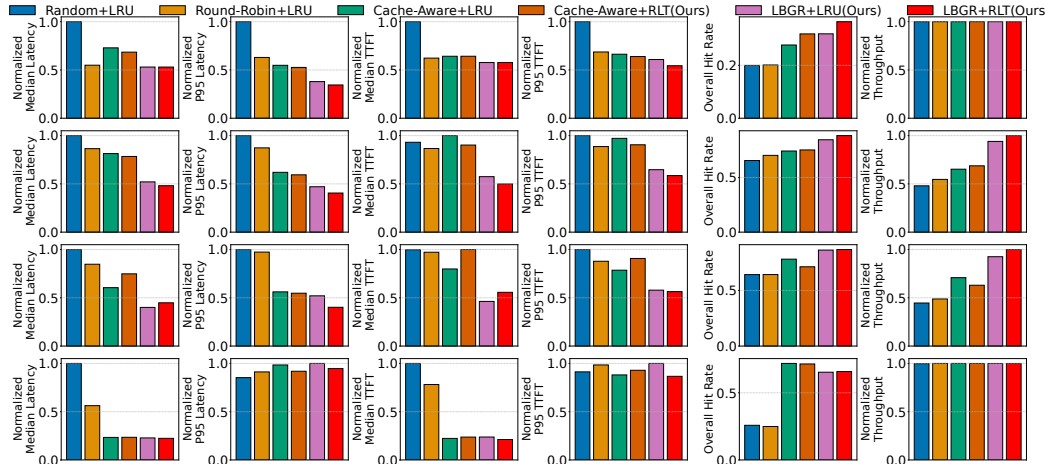

Figure 9: Results on Mixtral-8×7B-Instruct-v0.1 (Latency, TTFT, and Throughput normalized for comparison). Rows correspond to: GSP (top), ShareGPT (second), UltraChat (third), and Loogle (bottom). For the first four metrics, lower is better; for the last two, higher is better. Our algorithms consistently outperform all baselines across all benchmarks and metrics.

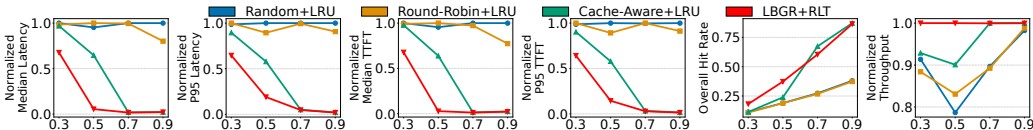

Figure 10: Results on Llama-3.1-8B-Instruct under GSP benchmark with varying shared prefix ratio.

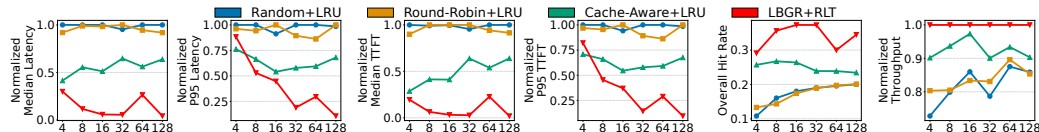

Figure 11: Results on Llama-3.1-8B-Instruct under GSP benchmark with varying number of queries, where the $x$-axis denotes the number of queries per shared-prefix group.

baselines across almost all group sizes. As the number of queries per group increases, the performance gap between LBGR+RLT and the baselines continues to widen. These results demonstrate the strong robustness and adaptability of our algorithms under diverse prefix-sharing conditions.

**Number of Output Tokens.** Figure 12 presents the performance of our method, LBGR+RLT, under varying maximum generated output lengths, ranging from 4 to 128 tokens. The results clearly show that LBGR+RLT outperforms all baselines across all metrics and output lengths, further demonstrating its strong performance under diverse output generation lengths.

**Maximum Concurrent Queries.** We vary the limit on the maximum number of concurrent queries per worker from 1 to 64. As shown in Figure 13, LBGR+RLT significantly outperforms all baselines across all settings, demonstrating its strong performance under varying batch sizes. Notably, Cache-Aware+RLT consistently outperforms Cache-Aware+LRU across all metrics and settings, which validates our theoretical analysis transferring in practice: Our eviction algorithm, RLT, achieves a significantly better competitive ratio compared to L-LRU in both the single-query and batch processing settings.

**Performance of RLT on Single Worker.** We evaluate the performance of RLT on a single worker under the worst-case query arrival order in the GSP benchmark, using 64 shared-prefix groups with 32 queries each. Table 3 reports the normalized comparison between L-LRU and RLT, showing that RLT reduces median latency and median TTFT by **45%** and **47%**, respectively. It also improves tail

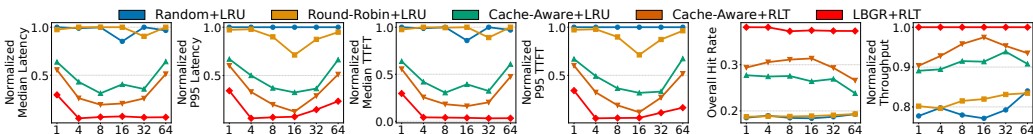

Figure 12: Results on Llama-3.1-8B-Instruct under GSP benchmark with varying maximum output token lengths.

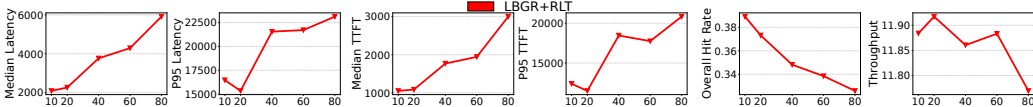

Figure 13: Results on Llama-3.1-8B-Instruct under GSP benchmark with varying limits on the maximum number of concurrent queries processed by each worker.

Figure 14: Ablation results on Llama-3.1-8B-Instruct under GSP benchmark with varying decay interval $\Delta t$ from 10ms to 80ms.

Table 3: Comparison of RLT and L-LRU on the single worker under round-robin query arrivals on the GSP benchmark.

| Algorithm | Normalized P50 Latency | Normalized P95 Latency | Normalized P50 TTFT | Normalized P95 TTFT | Hit Rate | Normalized Throughput |
|---|---|---|---|---|---|---|
| L-LRU | 1.0 | 1.0 | 1.0 | 1.0 | 6.06 % | 0.62 |
| **RLT** | **0.55** | **0.53** | **0.55** | **0.54** | **41.93%** | **1.0** |

performance, reducing P95 latency and P95 TTFT by **45%** and **46%**. Notably, RLT significantly boosts cache hit rate, achieving a **6.92×** higher hit rate and a **77.4%** increase in throughput over L-LRU.

**Decay Interval $\Delta t$.** To investigate the impact of the decay factor used in LBGR, we vary the decay interval $\Delta t$ from 10ms to 80ms. As shown in Figure 14, increasing $\Delta t$ leads to a drop in performance. This result highlights the importance of the background decay thread: applying timely queue load decay enables a more accurate reflection of the current query load state for each worker. Without decay, or under a weak decay setting (i.e., a large $\Delta t$), the estimated queue load becomes stale and less responsive to recent traffic, which degrades routing decisions and overall performance.

## C  THEORETICAL PROOFS

### C.1  COMPETITIVE RATIO OF L-LRU

**Lemma 1.** *In RadixAttention, the amortized number of misses incurred by OPT in a phase $\P_v$ with $c$ clean tokens is lower bounded by $\max\{c/2, 1\}$.*

*Proof.* We denote by $d$ the number of tokens in the cache of OPT that do not appear in the cache of L-LRU at the beginning of phase $\P_v$. Similarly, let $e$ denote the number of tokens in the cache of OPT that do not appear in the cache of L-LRU at the end of phase $\P_v$.

By definition, there are $c$ clean tokens in $\P_v$, each of which is not present in the cache of L-LRU at the start of the phase. Thus, each clean token must cause a cache miss for L-LRU.

Now, consider the overlap between the caches of OPT and L-LRU at the start and end of $\P_v$.

**At the start of $\P_v$.** There are $d$ tokens in OPT's cache that are not in L-LRU's cache. Therefore, at least $c - d$ of the clean tokens are also not present in OPT's cache, which means OPT incurs at least $c - d$ misses during this phase.

**At the end of $\P_v$.** There are $e$ tokens in OPT's cache that are not in L-LRU's cache. Two possible types of tokens may remain in L-LRU's cache: (i) *ghost* tokens that in the first path $\Gamma_j$ processed in $\P_v$ that were not accessed at all during $\P_v$ but remained in the cache, and (ii) new tokens accessed during $\P_v$. The occurrence of the first type is because the phase boundary may split the first path $\Gamma_j$ between $\P_{v-1}$ and $\P_v$. Due to the leaf-based eviction behavior of L-LRU, the prefix of $\Gamma_j$ (in phase $\P_{v-1}$) may still exist in the cache even if not accessed in $\P_v$, while its suffix (in $\P_v$) will be evicted first, as evictions target leaf tokens first. Since the entire path $\Gamma_j$ must exist in the cache for both OPT and L-LRU after being processed, we denote $a$ as the number of the prefix tokens of $\Gamma_j$, and $b$ as the number of its suffix tokens (new tokens).

Note that the ghost tokens can only exist in the first path $\Gamma_j$ at the end of $\P_v$. Suppose that $u$ ghost tokens from paths other than $\Gamma_j$ exist in the cache. Then L-LRU will only be able to hold $B_i - a - u$ new tokens, requiring space for $a + u$ more tokens. Since ghost tokens are least recently used, they will be evicted first to make room for the new tokens. As a result, L-LRU will evict $u$ such tokens from other paths and $a$ from $\Gamma_j$, leaving only $b$ of $\Gamma_j$ (either ghost or new tokens) in the cache.

We then analyze the cache state of OPT. The $e$ different tokens in OPT's cache but not in L-LRU's fall into three types: (i) $w$ tokens that never exist in the cache of L-LRU from the start to the end of $\P_v$; (2) $y$ new tokens from $\Gamma_j$ that were evicted by L-LRU; and (3) $z$ ghost tokens from $\Gamma_j$ that were evicted by L-LRU. Thus, we have $e = w + y + z$. Consider the following three cases:

- **Case1: When $y > 0$.** This indicates that OPT contains the full $a$ ghost tokens. Hence, OPT holds $w + a$ tokens that never appear during $\P_v$, leaving only $B_i - (w + a)$ slots to process $B_i$ new tokens. Therefore, it must occur at least $w + a$ misses. Since at most $a$ tokens from $\Gamma_j$ are evicted by L-LRU, we have $y + z \leq a$. Thus, OPT must occur at least $w + a \geq w + y + z \geq e$ misses.

- **Case 2: When $y = 0$ and $z > 0$.** Since $z > 0$, it indicates that $a > b$ and there are only $b$ ghost tokens left in the cache of L-LRU (i.e., $b = a + b - a$), and therefore, OPT holds a total $b + z$ ghost tokens in the cache. Therefore, it ocurrs at least $w + b + z \geq w + y + z = w + z = e$ misses.

- **Case 3: When $y = 0$ and $z = 0$.** Then, we have $w = e$, and OPT occurs at least $e$ misses.

Combining the above, the number of misses incurred by OPT in phase $\P_v$ is at least $\max\{c - d, e\} \geq \frac{1}{2}(c - d + e)$. Summing over all phases, the amortized number of misses incurred by OPT in a phase $\P_v$ is at least $c/2$. Finally, since OPT must incur at least one miss per phase (because a cache of size $B_i$ cannot hold all $B_i + 1$ different tokens), the amortized number of misses per phase is at least $\max\{c/2, 1\}$. $\qquad\square$

**Lemma 2.** *For any phase $\P_v$ with $c$ clean tokens, the number of misses incurred by L-LRU under single-query processing is at most $B_i - \mathcal{L} + c$, assuming no old token reappears after eviction.*

*Proof.* If $c \geq \mathcal{L}$, then $B_i - \mathcal{L} + c \geq B_i$. Since misses only occur for new tokens and there are at most $B_i$ new tokens in a phase, the maximum number of misses for L-LRU in this case is $B_i$, which is bounded by $B_i - \mathcal{L} + c$.

If $c < \mathcal{L}$, we analyze the maximum number of new tokens that can incur misses. Consider the first path $\Gamma_j$ after processing the initial clean tokens $c_1$.

As shown in Figure 15 (a), if $\Gamma_j$ is completed during the phase, there are at least $\mathcal{L} - c$ new tokens that must be processed that do not incur any misses. This is because $\Gamma_j$ consists of three components: (i) tokens already stored in the cache, (ii) tokens that might be evicted while loading the initial clean tokens (at most $c_1$), and (iii) its own clean tokens ($c_2$). The second and third parts (which will incur misses) together account for at most $c$ tokens, and the first part, at least $\mathcal{L} - c$ tokens, will not result in additional misses. Then, the total number of misses of L-LRU is bounded by $B_i - \mathcal{L} + c$.

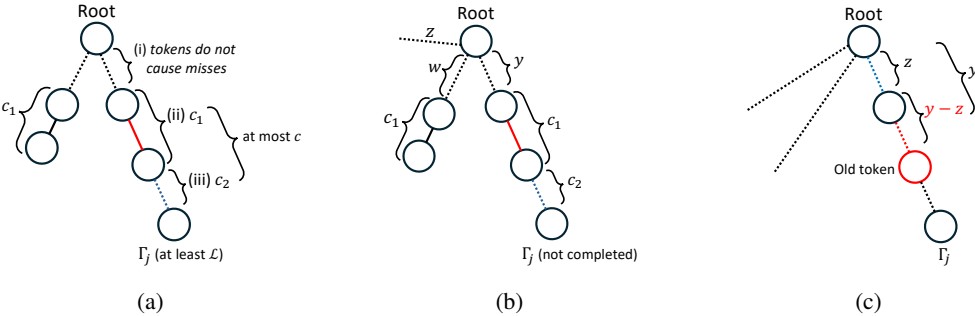

Figure 15: Visualizing key ideas in theoretical proofs

If $\Gamma_j$ is not completed during the phase, as shown in Figure 15 (b), the maximal number of misses for L-LRU is $y + c_1 + c_2 = y + c$, where $y$ is the number of prefix tokens of $\Gamma_j$ that initially exist in the cache but are evicted when loading the initial clean tokens. Assume there are $w$ prefix tokens from the initial clean tokens, and $z$ other distinct tokens remaining in the cache that are on the same paths as the initial clean tokens and $\Gamma_j$. Then, we have

$$y \le B_i - z - (w + c_1) \tag{7}$$

where $c_1$ is the initial clean tokens. Therefore, the total number of misses can be bounded as

$$y + c \le B_i - z - (w + c_1) + c \le B_i - z + c - \mathcal{L} \le B_i + c - \mathcal{L} \tag{8}$$

where it follows from $z \ge 0$ and $w + c_1 \ge \mathcal{L}$.

Therefore, the maximal total number of misses of L-LRU is bounded by $B_i - \mathcal{L} + c$ assuming that no old token reappears after being evicted. □

**Lemma 3.** *Under single-query processing in L-LRU, no old token reappears in any $\P_v$ after eviction.*

*Proof.* Let us analyze the first occurrence of such a case for an old token. This token can be either a leaf or an internal token in its path $\Gamma_j$. As shown in Figure 15 (c), assume this old token has $y$ prefix tokens, and $z$ tokens have been accessed among these $y$ prefix tokens during this phase. This indicates that at least $B_i - y + z$ new tokens must have been accessed by the time the old token is evicted. To see this, note that if this token is being evicted, all other leaf tokens currently in the cache must have been accessed more recently. Furthermore, because tokens within the same path are processed consecutively, the prefix tokens of these other leaf tokens are also more recently accessed.

Since $z$ of the old token's prefix tokens have already been visited, at least $B_i - y + z$ new tokens have been accessed at the time the old token is evicted. When revisiting this old token, its $y$ prefix tokens must be accessed again, of which $z$ have already been processed. Thus, there are $y - z$ new tokens to be assessed before the revisit. In total, this means $B_i - y + z + y - z = B_i$ tokens are accessed before the old token is revisited. Therefore, the phase $\P_v$ ends here, and case (2) cannot occur. □

**Theorem 4** (Single-Query). *Under single-query processing setting, the competitive ratio of L-LRU in RadixAttention on worker $m_i$ with cache capacity $B_i$ is upper bounded by $(B_i - \mathcal{L} + 2)$ and lower bounded by $(B_i - \mathcal{L} + 1)$, where $\mathcal{L}$ denotes the minimum length over all $\Gamma_j \in \widetilde{Q}$.*

*Proof.* For the first phase $\P_1$, since the cache is initially empty, OPT behaves the same as L-LRU. For any subsequent phase $\P_v$ ($v > 1$), we analyze the following two bounds:

**Upper Bound.** First, note that OPT incurs at least $\max\{c/2, 1\}$ misses in any phase $\P_v$ if there are $c$ clean tokens in that phase, according to Lemma 1. Furthermore, by the definition of a phase, where each phase consists of $B_i$ distinct tokens, the start of each phase is always associated with a path that ends with the clean tokens.

Next, let us analyze the maximum number of misses that L-LRU can incur for a given phase $\P_v$ with $c$ clean tokens. A miss can occur in one of two cases: (1) a new token appears, or (2) an old token appears after being evicted.

According to Lemma 3, case (2) cannot occur during a phase, so according to Lemma 2, the total number of misses of L-LRU is bounded by $B_i - \mathcal{L} + c$.

Therefore, the upper bound of the competitive ratio of L-LRU is bounded by

$$\max_c \left\{ \frac{B_i - \mathcal{L} + c}{\max\{c/2, 1\}} \right\} = B_i - \mathcal{L} + 2 \tag{9}$$

**Lower Bound.** We construct a set of paths that share the same $\mathcal{L} - 1$ prefix but with only one leaf token different. We then repeatedly query $\Gamma_1, \Gamma_2, \ldots, \Gamma_{B_i - \mathcal{L} + 2}$ in a loop. In this scenario, OPT can only incur 1 miss, while L-LRU incurs $B_i - \mathcal{L} + 1$ misses in each phase. Therefore, the lower bound for the competitive ratio of L-LRU is $B_i - \mathcal{L} + 1$. □

Based on the proof of single-query processing, we introduce the proof of batch processing.

**Theorem 5** (Batch). *Consider the continuous batch setting with batch size $\beta$. Let $\mathcal{L}_{max}$ and $\mathcal{L}$ denote the maximum and minimum lengths over all $\Gamma_j \in \widetilde{Q}$. If $\beta\mathcal{L}_{max} \leq B_i$, where $B_i$ is the cache capacity of worker $m_i$, and all queries in a batch are distinct, then, the competitive ratio of L-LRU in RadixAttention is upper bounded by $(B_i - \mathcal{L} - \beta + 3)$ and lower bounded by $(B_i - \mathcal{L} - \beta + 2)$.*

*Proof.* We assume that $\beta\mathcal{L}_{max} \leq B_i$, which ensures that the cache can always hold $\beta$ queries simultaneously at any point during generation. Based on this, there will be no abnormal evictions, such as evicting the KV values of tokens that belong to queries currently being generated. We now consider the following two bounds.

**Upper Bound.** According to Theorem 4, consider the scenario where case (2) does not occur. Let us examine the first complete batch $S_j$ that follows the initial clean tokens within any phase $\P_v$. Following the same analysis, if $c \geq \mathcal{L} + \beta - 1$, we have $B_i - \mathcal{L} - \beta + c + 1 \geq B_i$, so the total number of misses is bounded by $B_i - \mathcal{L} - \beta + c + 1$. If $c < \mathcal{L} + \beta - 1$, we analyze the maximum number of new tokens that can incur misses. If $S_j$ is completed during the phase, there are at least $\mathcal{L} + \beta - 1 - c$ new tokens that must be processed but do not incur any misses. This is because the tokens in $S_j$ can be categorized into three components: (i) tokens already present in the cache, (ii) tokens that may be evicted when loading the initial clean tokens and will incur misses when revisited, and (iii) its own clean tokens that incur misses. The first part, which does not contribute to additional misses, contains at least $\mathcal{L} + \beta - 1 - c$ tokens. This follows from the fact that the minimal number of distinct tokens in a batch is $\mathcal{L} + \beta - 1$. Therefore, the total number of misses of L-LRU is bounded by $B_i - \mathcal{L} - \beta + 1 + c$.

If $S_j$ is not completed during the phase, that is, if any queries in $S_j$ are not finished, then the batch is considered unfinished. In this case, the maximal number of misses can be represented as $y + c$, where $y$ is the number of prefix tokens of $S_j$ that initially reside in the cache but are evicted when loading the initial clean tokens. This is because, in the continuous batch setting, new queries can be regarded as incurring misses and as a continuation of those that have already been completed. Let $w$ be the number of prefix tokens of the initial clean tokens, and let $z$ denote the number of other distinct tokens remaining in the cache that are on the same paths as the initial clean tokens and $S_j$. Thus, the total number of misses is bounded as follows:

$$y + c \leq B_i - z - (w + c_1) + c \leq B_i - z + c - \mathcal{L} + 1 - \beta \leq B_i + c - \mathcal{L} + 1 - \beta \tag{10}$$

where the inequalities follow from $z \geq 0$ and $w + c_1 \geq \mathcal{L} - 1 + \beta$.

Therefore, when case (2) does not occur, the total number of misses of L-LRU is bounded by $B_i - \mathcal{L} - \beta + 1 + c$.

Now, consider the first time situation (2) occurs, where a miss is caused by revisiting an old token that has been evicted. It is possible that there are multiple old tokens involved, with at most $\beta$ different old tokens evicted and revisited simultaneously. Let us focus on the oldest among them, i.e., the old token that was visited earliest during the phase. Applying the same analysis as in Theorem 4 to this oldest token, we conclude that, upon revisiting, at least $B_i$ new tokens must have been accessed since it was evicted. Therefore, the phase $\P_v$ ends here, and case (2) cannot occur again within the same phase.

For OPT, it incurs at least $\max\{c/2, 1\}$ misses, according to Lemma 1. Thus, the competitive ratio of L-LRU is upper-bounded by

$$\max_c \left\{ \frac{B_i - \mathcal{L} - \beta + c + 1}{\max\{c/2, 1\}} \right\} = B_i - \mathcal{L} - \beta + 3 \tag{11}$$

**Lower Bound.** We construct $B_i - \mathcal{L} + 2$ paths that share the same $\mathcal{L} - 1$ tokens prefix but with only one leaf token different, $\{\Gamma_1, \Gamma_2, \ldots, \Gamma_{B_i - \mathcal{L} + 2}\}$. Furthermore, we construct $B_i - \mathcal{L} - \beta + 3$ batches, where $u$-th batch contains paths $\{\Gamma_1, \Gamma_2, \ldots, \Gamma_{\beta-1}, \Gamma_{u+\beta-1}\}$. We can do the following querying: $1, 2, \ldots, B_i - \mathcal{L} - \beta + 3$ batch in a loop. Therefore, for each phase, OPT can only incur 1 miss, whereas the L-LRU incurs $B_i - \mathcal{L} - \beta + 2$ misses per phase. Therefore, the lower bound for the competitive ratio of L-LRU is $B_i - \mathcal{L} - \beta + 2$. $\qquad\square$

**Discussion of the assumption $\beta\mathcal{L}_{max} \leq B_i$.** We assume that $\beta\mathcal{L}_{max} \leq B_i$, which ensures that the cache can always hold $\beta$ queries simultaneously at any point during running and thus prevents abnormal evictions (e.g., evicting KV entries for queries that are still being generated). This condition models the capacity constraint in practical LLM systems: GPU memory is limited, so system operators typically choose a maximum batch size $\beta$ and context length $\mathcal{L}_{max}$ such that their product does not exceed the available KV cache capacity $B_i$. If instead $\beta\mathcal{L}_{max} > B_i$ holds persistently, any policy that forms such batches would eventually be forced to evict KV entries belonging to in-progress queries, leading to degenerate cache behavior and a high risk of out-of-memory failures. Therefore, our analysis focuses on the well-provisioned regime where $\beta\mathcal{L}_{max} \leq B_i$, which accurately reflects real-world configurations and is precisely the setting where the choice of eviction policy has a meaningful impact.

We assume that queries in a batch are distinct to obtain a meaningful worst-case analysis for the batched setting. Without any diversity constraint, one can always construct worst-case batches by repeating a single query path $\beta$ times In this case, the behavior reduces to the single-query setting and the upper bound of competitive ratio becomes $B_i - \mathcal{L} + 2$. In Theorem 5, it considers all queries in the batch are different, which ensures the batch must contain at least $L + \beta - 1$ distinct tokens. This exactly leads to the $-\beta$ term in the final upper bound $B_i - \mathcal{L} - \beta + 3$. More generally, one can consider that each batch contains at least $d$ distinct queries, then the same analysis gives an upper bound of $B_i - \mathcal{L} - d + 3$ with $d \leq \beta$, which is weaker than the bound in Theorem 5. In deployed LLM services, batches are formed from requests arriving within a short time window, typically from different users, so batch contents are often diverse and can be reasonably approximated as distinct. Furthermore, due to the stochastic nature of LLM generation, even identical inputs may lead to different output paths, further increasing diversity within a batch. Thus, we focus on the representative high-diversity case $d = \beta$, where all queries in a batch are distinct.

## C.2 COMPETITIVE RATIO OF RLT

**Theorem 6** (Single-Query). *RLT is $\Theta(\log(B_i - \mathcal{L}))$-competitive on worker $m_i$ with cache capacity $B_i$ under single-query processing setting, where $\mathcal{L}$ is the minimal length over all $\Gamma_j \in \widetilde{Q}$.*

*Proof.* First, by Lemma 1, any algorithm, including OPT, incurs an amortized number of misses of at least $\max\{c/2, 1\}$ per phase. We now bound the expected number of misses incurred by RLT in a given phase. Following the analysis in (Fiat et al., 1991), we partition the $B_i$ new tokens into two groups: (1) clean tokens, which are not present in RLT's cache at the start of phase $\P_v$, and (2) stale tokens, which are present in the cache at the beginning of $\P_v$ but may be evicted during the phase.

In this setting, a miss can occur either when accessing a clean token or when accessing a stale token that has been evicted earlier in the phase. To maximize the number of misses, an adversarial request sequence first accesses all clean tokens, causing the eviction of some stale tokens. Subsequent requests to these evicted stale tokens could then incur the maximal number of additional misses. This is because accessing all clean tokens first increases the chance that the stale tokens will be evicted, therefore increasing the number of misses.

The number of misses incurred by clean tokens is clearly $c$. For each stale token, the expected cost of a miss equals the probability that it has been evicted by the time it is accessed. This probability is maximized at $c/s$, where $c$ is the number of clean tokens (i.e., the maximum number of evicted stale

tokens at any moment during the phase), and $s$ is the number of stale tokens that have not yet been accessed at that point.

According to Theorem 4, the total number of misses in a phase is upper bounded by $\min\{B_i - \mathcal{L} + c, B_i\}$. Therefore, the number of stale tokens that may incur a miss in RLT is at most $n = \min\{B_i - \mathcal{L} + c, B_i\}$. Hence, the total expected number of misses caused by stale tokens is upper bounded by:

$$\sum_{u=0}^{n-c-1} \frac{c}{n-u} = c(H_n - H_c) \tag{12}$$

where $H$ represents the Harmonic number.

Then, the expected number of misses incurred by RLT in a phase is upper bounded by $c + c(H_n - H_c)$. Accordingly, the competitive ratio of RLT is bounded by:

$$\max_c \left\{ \frac{c + c(H_n - H_c)}{\max\{c/2, 1\}} \right\} \tag{13}$$

Assuming $\mathcal{L} \geq 2$, which holds in most practical LLM-serving scenarios, the above expression yields a competitive ratio of:

$$\Theta\left(\log(B_i - \mathcal{L} + 2)\right) = \Theta\left(\log(B_i - \mathcal{L})\right) \tag{14}$$

Therefore, RLT is $\Theta(\log(B_i - \mathcal{L}))$-competitive on worker $i$ with cache capacity $B_i$ under single-query processing. $\square$

**Corollary 7** (Batch). *RLT is $\Theta(\log(B_i - \mathcal{L} - \beta))$-competitive on worker $m_i$ with capacity $B_i$ under continuous batching setting, where $\mathcal{L}$ is the minimal length over all $\Gamma_j \in \widetilde{Q}$, and $\beta$ is the batchsize.*

*Proof.* Following the same analysis in Theorem 6, we have that the total number of misses in a phase is upper bounded by $\min\{B_i - \mathcal{L} - \beta + c + 1, B_i\}$. Therefore, $n = \min\{B_i - \mathcal{L} - \beta + c + 1, B_i\}$ gives the maximum number of stale tokens that may result in misses in RLT. Accordingly, the total expected number of misses caused by stale tokens is also upper-bounded by:

$$\sum_{u=0}^{n-c-1} \frac{c}{n-u} = c(H_n - H_c) \tag{15}$$

Then, the competitive ratio of RLT is similarly bounded by:

$$\max_c \left\{ \frac{c + c(H_n - H_c)}{\max\{c/2, 1\}} \right\} \tag{16}$$

With $\mathcal{L} \geq 2$, we have the following competitive ratio:

$$\Theta\left(\log(B_i - \mathcal{L} - \beta + 3)\right) = \Theta\left(\log(B_i - \mathcal{L} - \beta)\right) \tag{17}$$

Therefore, RLT is $\Theta(\log(B_i - \mathcal{L} - \beta))$-competitive on worker $i$ with cache capacity $B_i$ under continuous batching generation. $\square$

Next, we show that no dependent algorithm can achieve a competitive ratio better than $\Theta(\log(B_i - \mathcal{L}))$ in the single-query processing setting, and $\Theta(\log(B_i - \mathcal{L} - \beta))$ in the continuous batching setting.

**Theorem 8** (Single-Query). *No randomized eviction algorithm can achieve a competitive ratio better than $\Theta(\log(B_i - \mathcal{L}))$ on worker $m_i$ with cache capability of $B_i$ in the single-query processing setting, where $\mathcal{L}$ denotes the minimal length over all $\Gamma_j \in \widetilde{Q}$.*

*Proof.* To establish the lower bound, we apply Yao's principle (Yao, 1977). Specifically, we first show that there exists a distribution of input query paths $\{x_1, x_2, \ldots, x_n\}$ such that any *deterministic* eviction algorithm $D$ incurs a competitive ratio of at least $H_{B_i - \mathcal{L} + 2}$.

The construction is as follows: each path shares a common prefix of length $\mathcal{L} - 1$ but differs in the last tail token. We sample each query path $\Gamma_j$ uniformly at random from the index set $j \in [B_i - \mathcal{L} + 2]$,

where $[B_i - \mathcal{L} + 2] = \{1, \ldots, B_i - \mathcal{L} + 2\}$. Under this distribution, the expected number of cache misses incurred by any $D$ is:

$$\mathbb{E}[Miss_D] = B_i + \frac{n - (B_i - \mathcal{L} + 1)}{B_i - \mathcal{L} + 2} \geq \frac{n}{B_i - \mathcal{L} + 2} \tag{18}$$

where the first $B_i$ terms account for the initial misses for $B_i$ tokens, incurred by the $(B_i - \mathcal{L} + 1)$ distinct query paths, the second term corresponds to subsequent misses, where each of the remaining $n - (B_i - \mathcal{L} + 1)$ query paths has a probability $\frac{1}{B_i - \mathcal{L} + 2}$ of incurring a miss.

Next, by Lemma 1, we have a lower bound on the number of misses incurred by OPT:

$$Miss_{\text{OPT}} \geq V(\max\{c/2, 1\}) \geq V \tag{19}$$

where $V$ is the number of phases in the query sequence. When $V$ is a random variable, it follows that $\mathbb{E}[Miss_{\text{OPT}}] \geq \mathbb{E}[V]$.

Since OPT always evicts the token whose next use is furthest in the future, it will evict the tail token furthest in the future. Suppose such a token is evicted at time $w$ and reappears at time $y$. Then, there will be no cache misses between $w$ and $y$. This is because there are only $B_i - \mathcal{L} + 2$ distinct query paths, and the cache can hold at most $B_i - \mathcal{L} + 1$ of them, which means only the evicted path is temporarily excluded. Furthermore, any other tail token must appear at least once between $w$ and $y$; otherwise, it would have been a better candidate for eviction due to its more distant reuse, which contradicts the eviction rule of OPT. Thus, the expected number of queries between two consecutive misses (i.e., between $w$ and $y$) follows the structure of the classical coupon collector's problem:

$$\mathbb{E}[y - w] = (B_i - \mathcal{L} + 2) \sum_{u=1}^{B_i - \mathcal{L} + 2} \frac{1}{u} = (B_i - \mathcal{L} + 2) H_{B_i - \mathcal{L} + 2} \tag{20}$$

which leads to

$$\mathbb{E}[Miss_{\text{OPT}}] = \frac{n}{(B_i - \mathcal{L} + 2) H_{B_i - \mathcal{L} + 2}} \tag{21}$$

Therefore, the competitive ratio of any deterministic algorithm $D$ is lower bounded by:

$$\frac{\mathbb{E}[Miss_D]}{\mathbb{E}[Miss_{\text{OPT}}]} = H_{B_i - \mathcal{L} + 2} \tag{22}$$

Finally, we leverage Yao's principle to establish a lower bound for the competitive ratio of any randomized algorithm. Let $X^n$ denote the random variable representing the input query paths $\{x_j\}_{j=1}^n$. Then, we have

$$\mathbb{E}_{X^n}[Miss_D(X^n)] \geq H_{B_i - \mathcal{L} + 2} \mathbb{E}_{X^n}[Miss_{\text{OPT}}(X^n)] \tag{23}$$

We assume $\mathcal{D}$ is the distribution over the deterministic algorithms. Taking the expectation over $D \sim \mathcal{D}$, we have

$$\mathbb{E}_{X^n} \mathbb{E}_{\mathcal{D}}[Miss_{\mathcal{D}}(X^n)] \geq H_{B_i - \mathcal{L} + 2} \mathbb{E}_{X^n}[Miss_{\text{OPT}}(X^n)] \tag{24}$$

By the definition of expectation, it indicates that there exists a specific sequence of input paths $\{x_j^*\}_{j=1}^n$, such that

$$\mathbb{E}_{\mathcal{D}}[Miss_{\mathcal{D}}(\{x_j^*\})] \geq H_{B_i - \mathcal{L} + 2} Miss_{\text{OPT}}(\{x_j^*\}) \tag{25}$$

Therefore, for any randomized eviction algorithm, its competitive ratio under the single-query processing setting is lower bounded by:

$$H_{B_i - \mathcal{L} + 2} = \Theta\left(\log(B_i - \mathcal{L})\right) \tag{26}$$

$\square$

**Corollary 9** (Batch). *No randomized eviction algorithm can achieve a competitive ratio better than $\Theta(\log(B_i - \mathcal{L} - \beta))$ on worker $m_i$ with cache capability of $B_i$ in the continuous batching setting, where $\mathcal{L}$ denotes the minimal length over all $\Gamma_j \in \widetilde{Q}$ and $\beta$ is the batch size.*

*Proof.* Following the analysis in Theorem 8, we first construct $B_i - \mathcal{L} + 2$ distinct paths $\{\Gamma_1, \Gamma_2, \ldots, \Gamma_{B_i - \mathcal{L} + 2}\}$, each sharing the same $\mathcal{L} - 1$ tokens prefix but differing in their final (tail) token. Based on these paths, we define $B_i - \mathcal{L} - \beta + 3$ distinct batches, where the $u$-th batch consists of the paths $\{\Gamma_1, \Gamma_2, \ldots, \Gamma_{\beta-1}, \Gamma_{u+\beta-1}\}$. Now consider the given query batch sequence, $\{x_1, x_2, \ldots, x_n\}$, where each $x_i$ is a batch picked uniformly at random from the total $B_i - \mathcal{L} - \beta + 3$ batches. Then, the expected number of misses incurred by any deterministic algorithm $D$ is then lower bounded by:

$$\mathbb{E}[Miss_D] = B_i + \frac{n - (B_i - \mathcal{L} - \beta + 2)}{B_i - \mathcal{L} - \beta + 3} \geq \frac{n}{B_i - \mathcal{L} - \beta + 3} \tag{27}$$

where this follows from the fact that at least $B_i - \mathcal{L} - \beta + 2$ distinct batches are required to fill an initially empty cache.

According to Lemma 1 and Theorem 8, the expected number of misses incurred by OPT satisfies

$$E[Miss_{\text{OPT}}] \geq \mathbb{E}[V(\max\{c/2, 1\})] \geq \mathbb{E}[V] \tag{28}$$

where $V$ is the number of phases for the given queries.

Next, note that under the given query batches, OPT only evicts one token at a time. This is because each batch contains exactly one unique tail token from the last path $\Gamma_{u+\beta-1}$, and this setup mirrors the single-query setting analyzed in Theorem 8. Therefore, by applying the same analysis, the expected number of tokens between any two misses (say, between $w$ and $y$) follows:

$$\mathbb{E}[y - w] = (B_i - \mathcal{L} - \beta + 3) \sum_{u=1}^{B_i - \mathcal{L} - \beta + 3} \frac{1}{u} = (B_i - \mathcal{L} - \beta + 3) H_{B_i - \mathcal{L} - \beta + 3} \tag{29}$$

which leads to

$$\mathbb{E}[Miss_{\text{OPT}}] = \frac{n}{(B_i - \mathcal{L} - \beta + 3) H_{B_i - \mathcal{L} - \beta + 3}} \tag{30}$$

Therefore, the competitive ratio of any deterministic eviction algorithm $D$ in the continuous batching setting is lower bounded by:

$$\frac{\mathbb{E}[Miss_D]}{\mathbb{E}[Miss_{\text{OPT}}]} = H_{B_i - \mathcal{L} - \beta + 3} \tag{31}$$

Finally, by Yao's principle, we conclude that for any randomized eviction algorithm under the continuous batching setting, its competitive ratio is at least

$$H_{B_i - \mathcal{L} - \beta + 3} = \Theta\left(\log(B_i - \mathcal{L} - \beta)\right) \tag{32}$$

$\square$

# D EXTENDED RELATED WORK

**KV Cache Optimization.** The computational complexity of Large Language Models (LLMs) in token generation scales quadratically with sequence length (Jaillet et al., 2026; Ainslie et al., 2023; Vaswani et al., 2017). KV caching is a fundamental optimization that mitigates this complexity by storing the computed KV pairs for past tokens and reusing them for future queries (Kwon et al., 2023; Lee et al., 2024; Sheng et al., 2024). However, the limited memory capacity of KV caches remains a bottleneck for long-context generation, where storing the entire historical token context is impractical (Zhang et al., 2023; Xiao et al., 2024). Researchers, therefore, have explored memory reduction through quantization techniques that compress cached KV values (Lin et al., 2016; Wu et al., 2020; Zhou et al., 2018; Jiang & Agrawal, 2018). This involves converting the full-precision values (e.g., FP16) stored in the cache to lower-precision integer formats (e.g., INT8) (Yao et al., 2022; Sheng et al., 2023; Hooper et al., 2024). However, outlier KV values often lead to significant performance degradation during quantization (Dettmers et al., 2022; Xiao et al., 2023). This has motivated specialized techniques such as SmoothQuant (Xiao et al., 2023) and KVQuant (Hooper et al., 2024), which smooth or isolate outliers to improve quantization robustness.

**KV Cache Management.** Recent work on KV cache management falls into two complementary strands. Context-aware methods leverage model-driven signals (e.g., attention weights or token

importance) to score tokens, retaining only a fixed-budget subset and evicting the rest (Zhang et al., 2023; Xiao et al., 2024; Liu et al., 2023). Representative examples include H2O, which preserves Heavy-Hitters with high attention scores (Zhang et al., 2023), and StreamingLLM, which retains initial attention sink tokens that anchor model stability (Zhang et al., 2023; Xiao et al., 2024). While effective at reducing memory, these strategies risk performance degradation on tasks requiring high-fidelity recall, since pruning may inadvertently discard critical long-range context. Another strand is system-oriented and context-agnostic, focusing on improving KV-cache utilization through redesigned memory layouts and management interfaces (Kwon et al., 2023; Zheng et al., 2024). vLLM's PagedAttention virtualizes the cache into fixed-size pages to reduce fragmentation and support efficient sharing under tight GPU memory (Kwon et al., 2023), while RadixAttention uses a radix tree over token prefixes to enable prefix reuse and fine-grained allocation/eviction for high-throughput serving (Zheng et al., 2024). However, those systems rely primarily on LRU-based policy for KV cache eviction, which is fragile under adversarial or bursty query patterns and can yield negligible hit-rate improvements in the worst case. This reliance, coupled with the lack of formal analysis for ordered, prefix-sharing KV cache structures, reveals a gap between practical designs and theoretical understanding in KV cache eviction.

**KV Cache-Aware Load Balancing.** In multi-LLM serving, balancing queue load while preserving KV reuse gives rise to the problem of KV cache–aware load balancing (Sun et al., 2024; Zheng et al., 2024; Lee et al., 2024). Most systems tackle this with heuristics that trade off cache affinity against query load (NVIDIA, 2025; Zheng et al., 2024; llm d, 2025). SGLang, for example, employs a rule-based strategy that switches between highest-hit-rate routing and least-loaded routing based on a predefined load-balance threshold (Zheng et al., 2024). Other systems instead adopt a static linear scoring function that combines prefix-match benefits with current load to guide routing (NVIDIA, 2025; llm d, 2025). SkyLB focuses on decentralized deployments, employing per-region coordinators and a multi-region prefix trie to preserve KV locality across regions (Xia et al., 2025). While practical, these methods remain largely heuristic and lack formal modeling for the underlying KV cache-aware load balancing problem, leaving them vulnerable to suboptimal performance under dynamic query patterns.

## E    DISCUSSION ON GLOBAL RADIX TREE

In our implementation of LBGR, we directly use the existing global radix tree in SGLang to estimate the cache hit rate for each worker for an incoming query. This global radix tree caches queries at the character level, avoiding tokenization and significantly reducing runtime overhead. It maintains per-worker cache state, where each worker corresponds to a subtree consisting of multiple root-to-leaf character paths, and each path may be associated with multiple workers simultaneously. For a new query, it performs a longest-prefix match against the subtree of each worker and estimates the hit rate based on the character-level match. This estimate may deviate from the true cache hit rate based on the worker's token-level cache due to two factors: (i) a mismatch between character-level and token-level granularity, and (ii) staleness or inconsistency in the shared tree caused by concurrent updates. The same global radix tree is also used across all baselines in our experiments, including the state-of-the-art method. A detailed investigation of this structure and its associated biases is beyond the scope of this work and is left for future research.

## F    LLM USAGE

We used LLMs only for language polishing and grammar correction. All technical content, theoretical analysis, algorithm design, and experimental results were developed independently by the authors.

