# OpenReview forum: "Randomization Boosts KV Caching, Learning Balances Query Load: A Joint Perspective"
_ICLR.cc/2026/Conference — ICLR 2026 Poster_

### Official Review · Reviewer_yivC · 2025-10-31

**Soundness:** 3
**Presentation:** 3
**Contribution:** 3
**Rating:** 6
**Confidence:** 3

**Summary:**

This paper addresses the critical trade-off between KV cache eviction and query routing in multi-LLM serving systems. It introduces the first unified mathematical model to formalize this trade-off, identifies theoretical limitations of existing LRU-based eviction policies, and proposes two novel algorithms: RLT eviction and LBGR.

**Strengths:**

1. The unified mathematical model fills a critical gap by explicitly coupling local cache eviction dynamics with global load balancing, a connection that prior heuristic-based methods failed to formalize.
2. RLT’s randomized eviction mechanism is simple to implement, seems simple but works.
3. The evaluation covers a broad range of scenarios: 4 benchmarks (synthetic and real-world), 3 prefix-sharing settings, model sizes from 8B to 70B (dense and MoE architectures).

**Weaknesses:**

1. The theoretical analysis notes that L-LRU’s performance degrades with imbalanced query lengths, but the paper does not explicitly evaluate how RLT/LBGR perform across different query length distributions (e.g., heavy-tailed vs. uniform).
2. The ablation table (Table 1) reports "Average Eviction Time" and "Average Routing Time" but does not compare these to baselines (e.g., L-LRU’s eviction time vs. RLT’s).

**Questions:**

1. I am wondering how to run the code? seems the authors do not contain a readme.
2. The state-of-the-art baseline is "Cache-Aware+LRU" from SGLang, which switches between hit-rate and load-based routing using a fixed threshold. Have you compared LBGR to dynamic threshold-based routing (where the threshold adapts to query arrival patterns)?

---

> ### Author Response · Authors · 2025-11-18
> **Rebuttal (1/2)**
>
> We thank the reviewer for their positive assessment and constructive suggestions.
>
> **Q1: The theoretical analysis notes that L-LRU’s performance degrades with imbalanced query lengths, but the paper does not explicitly evaluate how RLT/LBGR perform across different query length distributions (e.g., heavy-tailed vs. uniform).**
>
> A: Thank you for the helpful suggestion. Since we model the KV cache as operating at the token level, our theoretical analysis (Theorem 5) focuses on the impact of query-length heterogeneity. Specifically, when there is a meaningful gap in token lengths between short and long queries, increasing this gap worsens the worst-case competitive ratio and can lead to degraded performance for L-LRU. This worst-case analysis is agnostic to query arrival patterns or length distributions. To evaluate this empirically, we have already incorporated query length variability in the GSP benchmark by using a diverse set of query lengths: {512, 1024, 2048, 4096, 8192}. The results in Table 2 (formerly Table 1 in the original submission) show that, under this imbalanced-length setting, our methods significantly outperform the SOTA baseline. These results highlight the robustness of our algorithms in realistic and challenging scenarios involving non-uniform query lengths.
>
>
> To further investigate the effect of query-length distributions on our methods, we evaluate two GSP configurations on Llama-3.1-70B-Instruct: (i) a uniform-length setting, where all queries are 2048 tokens long, and (ii) a heavy-tailed setting, where 80% of queries have length 2048 and the remaining 20% have length 8192. The results in the following two tables show that our method outperforms the strongest baseline across all metrics, even in the uniform-length case where length heterogeneity is absent. In the heavy-tailed setting, this advantage becomes noticeably larger: our method reduces median latency and TTFT by **2.49×** and **3.68×**, respectively, over Cache-Aware+L-LRU, while also improving hit rate and throughput. These results align with our theoretical analysis of query-length imbalance and demonstrate that our method remains effective and robust under both uniform and highly skewed query-length distributions.
>
>
> **Table R1. Performance (ms) of LBGR+RLT under a uniform query-length distribution.**
> | Method    | P50 Latency | P95 Latency |P50 TTFT |P95 TTFT| Hit Rate| Throughput |
> |----------------|--------|-------|---------|------|------|------|
> | Cache-Aware+L-LRU| 411.40 | 1131.12 | 291.12 | 699.30 | 35.10% | 11.95|
> | **LBGR+RLT**     | **334.40** | **668.26**   | **204.74** | **431.94** | **41.08%**| **11.97**|
>
>
> **Table R2. Performance (ms) of LBGR+RLT under a heavy-tailed query-length distribution.**
> | Method    | P50 Latency | P95 Latency |P50 TTFT |P95 TTFT| Hit Rate| Throughput |
> |----------------|--------|-------|---------|------|------|------|
> | Cache-Aware+L-LRU| 29834.11 | 58217.23 | 27010.00 | 56303.01 | 24.87% | 10.38|
> | **LBGR+RLT**    | **11999.09** | **29413.14**   | **7321.59** | **27576.60** | **34.12%**| **11.54**|

---

> ### Author Response · Authors · 2025-11-18
> **Rebuttal (2/2)**
>
> **Q2: The ablation table (Table 1) reports "Average Eviction Time" and "Average Routing Time" but does not compare these to baselines (e.g., L-LRU’s eviction time vs. RLT’s).**
>
> A: Thank you for the helpful suggestion. In Table 2 (formerly Table 1 in the original submission), we report both the eviction and routing times for our methods as well as for the SOTA baseline (Cache-Aware routing with L-LRU, which we refer to as LRU for simplicity in the paper). Compared to the baseline, our method (RLT+LBGR) introduces only a small total runtime overhead of approximately 2 ms. Specifically, for eviction, RLT adds 0.58ms over L-LRU when using Cache-Aware routing, and 0.96ms when combined with LBGR, both of which are negligible relative to the overall end-to-end latency. For routing, LBGR adds 0.56ms over Cache-Aware routing when paired with L-LRU, and 0.94ms when paired with RLT, again resulting in minimal overhead. We also included this more detailed breakdown analysis in the evaluation section.
>
> **Q3: I am wondering how to run the code? seems the authors do not contain a readme.**
>
> A: Thank you for the valuable question. We have already included a README file in the anonymous repository, which provides detailed instructions for running the code. Please feel free to follow the steps outlined there and let us know if anything remains unclear.
>
> **Q4: The state-of-the-art baseline is "Cache-Aware+LRU" from SGLang, which switches between hit-rate and load-based routing using a fixed threshold. Have you compared LBGR to dynamic threshold-based routing (where the threshold adapts to query arrival patterns)?**
>
> A: Thanks for your helpful suggestion. Dynamic threshold-based routing is indeed an interesting and promising direction for this problem. However, to the best of our knowledge, no prior work has proposed or implemented a well-defined dynamic thresholding method for this problem.
>
> We hope that our responses and new results adequately address your concerns. If there are any remaining points we may have overlooked, we would be happy to continue the discussion.

---

> > ### Comment · Reviewer_yivC · 2025-11-26
> >
> > Thank you for the rebuttal. I have read all the responses and the revisions to the paper, which have further deepened my understanding of the work.

---

> > > ### Author Response · Authors · 2025-11-26
> > >
> > > Dear reviewer yivC,
> > >
> > > Thank you very much for carefully reading our responses and revisions, and for your follow-up comment! We hope that our rebuttal and the updated manuscript have adequately addressed your concerns. If there are any remaining issues, please feel free to let us know. If you feel that your concerns have been resolved, we would be grateful if you could consider raising your score.
> > >
> > >
> > > Best regards,
> > >
> > > The authors

---

### Official Review · Reviewer_hRYq · 2025-10-31

**Soundness:** 2
**Presentation:** 3
**Contribution:** 3
**Rating:** 6
**Confidence:** 2

**Summary:**

This paper presents the first unified theoretical model that couples KV cache eviction and query load balancing for multi-LLM serving. It identifies the fragility of LRU-based eviction under dynamic query arrivals and proposes two principled algorithms: RLT, a randomized eviction achieving logarithmic competitive ratio, and LBGR, a learning-based greedy routing method predicting end-to-end latency online.

**Strengths:**

1. This paper is clearly written, and the main problem is well-motivated from a practical LLM-serving perspective.
2. This paper provides a combination of a theoretical foundation and practical implementation.
3. I appreciate that the authors go beyond heuristic system designs and provide a theoretically grounded formulation together with competitive analysis for the cache eviction process.

**Weaknesses:**

1. As I understand, RLT may be affected by the random seed. It would be better to include an ablation study evaluating the stability of RLT under different random seed settings.
2. From Figure 6, it appears that the advantage of your method diminishes as the number of workers increases. Could you explain why the proposed approach cannot (or does not need to) scale to a larger number of workers?
3. Writing: It would be better to include a notation table in Section 3 to improve readability and help readers follow the theoretical formulation.

**Questions:**

Please see weaknesses.

---

> ### Author Response · Authors · 2025-11-18
> **Rebuttal**
>
> We appreciate the reviewer’s positive evaluation and helpful suggestions.
>
> **Q1: As I understand, RLT may be affected by the random seed. It would be better to include an ablation study evaluating the stability of RLT under different random seed settings.**
>
> A: Thanks for your valuable comments. To assess the stability of RLT, we conducted an ablation study on Llama-3.1-70B-Instruct over the GSP benchmark, varying the random seed over a diverse set: {0, 42, 123, 1000, 2048, 12345, 54321}. The following table reports mean±std across these seeds. The results show that LBGR+RLT consistently outperforms Cache-Aware+L-LRU across all evaluated seeds while maintaining stable performance and low variability across seeds. This demonstrates that RLT is robust to the choice of random seed.
>
> **Table R1. Performance (ms) of LBGR+RLT under varying random seeds.**
> | Method    | P50 Latency | P95 Latency |P50 TTFT |P95 TTFT| Hit Rate| Throughput |
> |----------------|--------|-------|---------|------|------|------|
> | Cache-Aware+L-LRU| 31049.58 | 56924.50 | 27971.49 | 52948.43 | 22.69% | 10.32|
> | **LBGR+RLT**     | **2,987.39±369.40** | **19,821.38±1093.00**   | **1,497.79±187.30** | **15995.63±1396.53** | **36.23%±0.41%**| **11.88±0.034**|
>
> **Q2: From Figure 6, it appears that the advantage of your method diminishes as the number of workers increases. Could you explain why the proposed approach cannot (or does not need to) scale to a larger number of workers?**
>
> A: Thank you for the detailed question. As shown in Figure 6, our method (LBGR+RLT) has a clear and consistent advantage over all baselines across all worker counts in terms of median and P95 latency, TTFT, and cache hit rate. For throughput, our method achieves higher values when the number of workers is relatively small (e.g., 2 or 4). As the number of workers increases ($\geq$ 6), the throughput of all methods converges to essentially the same value. This is expected: the request rate is fixed at 12 requests/s, so once the system has sufficient capacity, all methods are able to achieve the maximum throughput of 12 requests/s. In this lightly loaded scenario, there is inherently little room for further throughput improvement, even though LBGR+RLT still provides clearly better latency and cache hit rate. We have included this analysis in the revised draft.
>
>
> **Q3: Writing: It would be better to include a notation table in Section 3 to improve readability and help readers follow the theoretical formulation.**
>
> A: Thank you for the helpful suggestion. We have added a notation table in Section 3 to improve readability and help readers follow the theoretical formulation more easily.
>
> We believe the revisions and additional experiments address all of the concerns raised in your review. If we have missed anything, we would be glad to provide further clarification.

---

### Official Review · Reviewer_NRb7 · 2025-11-01

**Soundness:** 2
**Presentation:** 2
**Contribution:** 3
**Rating:** 6
**Confidence:** 4

**Summary:**

The paper proposes a unified model of KV cache–aware load balancing and introduces two algorithms that jointly improve cache hit rate and end-to-end latency. The authors show L-LRU has worst-case O(n) competitiveness while RLT achieves O(logn), and they report large empirical gains across four benchmarks and multiple model sizes.  ￼  ￼  ￼

**Strengths:**

- Clear problem framing that couples cache eviction with routing
- Strong empirical results across four benchmarks with higher hit rate and throughput￼

**Weaknesses:**

- Some assumptions are under-discussed (see questions)

**Questions:**

Thank you for the submission. I like the paper overall, the KVCache scheduling topic for load balancing is timely, the theoretical results are crisp, and the experiments are compelling. I especially appreciated the side-by-side algorithms and the clear latency decomposition in LBGR, which make the motivation and mechanics transparent. That said, a few descriptions felt a bit under-explained to me. I’d appreciate clarifications on the following:
- Theorem 5 assumes βL_{\max}\le B_i and that queries in a batch are distinct. How representative is this in deployed systems? What happens to the bounds or behavior when those assumptions are violated?
- How sensitive are results to the hyper-parameter settings choices across hardware/model sizes? Any guidance for setting them without tuning?
- Since you normalize many plots, it would help to include at least one table with absolute value so readers can reason about real-world SLOs.

---

> ### Author Response · Authors · 2025-11-18
> **Rebuttal (1/2)**
>
> We thank the reviewer for their positive feedback and constructive questions.
>
> **Q1: Theorem 5 assumes βL_{\max}\le B_i and that queries in a batch are distinct. How representative is this in deployed systems? What happens to the bounds or behavior when those assumptions are violated?**
>
> A: Thank you for your valuable comments. The condition $\beta\mathcal{L} _ {max} \leq B_i$ models the capacity constraint in practical LLM systems. It reflects the fact that GPU memory is limited, so system operators will typically select a proper maximum batch size $\beta$ and context length $\mathcal{L} _ {max}$ such that their product does not exceed the available KV cache capacity. If $\beta\mathcal{L} _ {max} > B_i$ holds persistently, then any policy that forms such batches would be forced to evict KV entries belonging to in-progress queries, resulting in degenerate cache behavior and a high risk of out-of-memory failures. Therefore, our analysis focuses on the well-provisioned regime where $\beta\mathcal{L} _ {max} \leq B_i$, which accurately reflects how real-world systems are configured and is precisely the setting where the choice of eviction policy has a meaningful impact.
>
> The reason for assuming that queries in a batch are distinct is to model a meaningful worst-case analysis for the batched setting. If we do not impose any diversity condition, in the worst-case analysis, one can always choose batches that consist of a single path repeated $\beta$ times. In this case, the behavior reduces to the single-query setting and the upper bound of competitive ratio becomes $B_i - \mathcal{L}+2$. In Theorem 5, it considers all queries in the batch are different, which ensures the batch must contain at least $\mathcal{L} + \beta - 1$ distinct tokens (two tokens are distinct if they appear at different positions along the path or have different values). This exactly leads to the $-\beta$ term in the final upper bound $B_i - \mathcal{L} - \beta + 3$. One can consider a more general model where, in each batch, there are at least $d$ distinct queries. Then, the same worst-case analysis suggests an upper bound of $B_i - \mathcal{L} - d + 3$ with $d \leq \beta$, which is weaker than the bound in Theorem 5. In deployed LLM services, batches are formed from requests arriving within a short time window, typically from different users, so batch contents are often diverse and can be reasonably approximated as distinct. Furthermore, due to the stochastic nature of LLM generation, even identical inputs may lead to different output paths, further increasing diversity within a batch. We therefore assume that the queries in a batch are distinct and directly analyze the representative high-diversity case $d = \beta$.
>
> We have also incorporated this detailed discussion into the revised manuscript.

---

> ### Author Response · Authors · 2025-11-18
> **Rebuttal (2/2)**
>
> **Q2: How sensitive are results to the hyper-parameter settings choices across hardware/model sizes? Any guidance for setting them without tuning?**
>
> A: Thanks for the valuable question. Our algorithm has three main hyperparameters: the token-hit time cost $\alpha_{\text{CACHED}}$, the token-miss time cost $\alpha_{\text{MISS}}$, and the decay interval $\Delta t$ (used in conjunction with a decay factor $\rho$). The decay strength is jointly determined by the pair $(\rho, \Delta t)$, so adjusting either has a similar effect. In our experiments, we fix $\rho$ and tune only $\Delta t$. For all experiments in the paper, we use a **single** set of hyperparameters, regardless of the model size or GPU type. Specifically, we set $\alpha_{\text{CACHED}} = 0$ms/1k tokens, $\alpha_{\text{MISS}} = 1000$ms/1k tokens, and $\Delta t = 20$ms for all runs, including Llama-3.1-8B-Instruct on NVIDIA L40, and Llama-3.1-70B-Instruct and Mixtral 8×7B on NVIDIA H200. We do not retune these parameters per model or hardware configuration, except in ablation studies where we explicitly analyze their effect.
>
> We already included an ablation study on the decay interval $\Delta t$ in Appendix B (Figure 14). The results show that the performance of our algorithms remains stable for small decay intervals (e.g., $\Delta t \in [10, 20]$ms). Furthermore, we vary the token-miss cost $\alpha_{\text{MISS}}$ on Llama-3.1-70B-Instruct over GSP benchmark from 200ms to 1400ms/1k tokens. The following table shows that LBGR+RLT reduces median latency and TTFT by up to **11.13×** and **20.00×**, respectively, over Cache-Aware+L-LRU, while also improving hit rate and throughput. The performance of LBGR+RLT remains stable for $\alpha_{\text{MISS}} \in [400, 1400]$, with its strong advantage over the SOTA baseline persisting.
>
> **Table R1. Performance (ms) of LBGR+RLT when varying $\alpha_{\text{MISS}}$.**
> | Method    | P50 Latency | P95 Latency |P50 TTFT |P95 TTFT| Hit Rate| Throughput |
> |----------------|--------|-------|---------|------|------|------|
> | Cache-Aware+L-LRU| 31049.58 | 56924.50 | 27971.49 | 52948.43 | 22.69% | 10.32|
> | LBGR+RLT ($\alpha_{\text{MISS}}=200$)     | 6475.21 | 21905.67   | 3148.65 | 19675.28 | 35.17%| 11.81 |
> | LBGR+RLT ($\alpha_{\text{MISS}}=400$)     | 3450.07 | 20792.63   | 1670.97| 16597.24 | **36.09%**| **11.89** |
> | LBGR+RLT ($\alpha_{\text{MISS}}=600$)     | 3582.41 | 20887.72   | 1763.63 | 18046.42 | 35.46%| **11.89**|
> | LBGR+RLT ($\alpha_{\text{MISS}}=800$)     | 3817.86 | 21357.47   | 1840.86 | 17761.47 | 35.41%| 11.88|
> | LBGR+RLT ($\alpha_{\text{MISS}}=1000$)     | 2859.72 | **19863.38**   | 1414.27 | **15222.20** | 35.83%| 11.82|
> | LBGR+RLT ($\alpha_{\text{MISS}}=1200$)     | **2788.27** | 20247.58   | **1398.92** | 16296.03 | 36.05%| 11.88|
> | LBGR+RLT ($\alpha_{\text{MISS}}=1400$)     | 3138.13 | 20192.63   | 1531.52 | 16190.72 | 35.96%| 11.84|
>
> **Guidance for setting parameters.** In our experiments, we use a simple strategy to choose the hyperparameters. Since the cost of a token hit is negligible compared to that of a token miss, we set $\alpha_{\text{CACHED}} = 0$ by default. To estimate the parameters for a target model on a specific device, we can first send a warm‑start batch of diverse queries (which can be synthetically generated) to the target model deployed on that device. For this batch, we record each query’s TTFT and input token count, as well as the batch’s total processing time. With these measurements, we estimate $\alpha_{\text{MISS}}$ as the average per-query, per-token time based on TTFT and input token counts. We then estimate $\Delta t$ from the total batch processing time with the given fixed decay factor $\rho$. In practice, one may also start from the paper’s default configuration when moving to new hardware or models. Since $\alpha_{\text{MISS}}$ denotes the time cost per missed token, slower models or less powerful GPUs naturally correspond to slightly larger values of $\alpha_{\text{MISS}}$ (and possibly a modestly larger $\Delta t$), while faster configurations would correspond to smaller values.
>
>
> **Q3: Since you normalize many plots, it would help to include at least one table with absolute value so readers can reason about real-world SLOs.**
>
> A: Thanks for your helpful suggestion. We have already included a detailed table (Table 2 in the current version of the paper) of absolute performance metrics. This table reports all metrics in absolute values and directly compares our methods with the state-of-the-art baseline.
>
>
> We believe we have addressed all of your concerns. If there are any remaining issues, we would be very happy to clarify them further.

---

### Official Review · Reviewer_HKW6 · 2025-11-01

**Soundness:** 3
**Presentation:** 4
**Contribution:** 3
**Rating:** 8
**Confidence:** 4

**Summary:**

The paper studies the concept of KV caching, which is fundamentally important for serving LLMs. The authors focus on providing a mathematical model to understand the interplay between KV cache hits and query load balancing. They provide theoretical results on the poor worst-case performance of algorithms like Leaf-LRU. They then provide two new algorithms: randomized leaf token (RLT) and learning-based greedy routing (LBGR). This new approach outperforms state

**Strengths:**

1. LLM KV cache managements and query routing is a very critical problem, and the paper does a good job of choosing an important problem to solve
2. The paper does quite a good job at piecing together the theoretical underpinnings of KV cache management, which makes the motivation of RLT and LBGR easy.
3. Section 3.1 does a great job at formalizing the notation and laying the groundwork for further sections. The lemmas are intuitive to understand
4. The experiments are quite extensive, covering large and small, dense and MoE models. The improvements across the board are strong

**Weaknesses:**

1. The improvements claims made in the intro should be qualified by model type and size, context length, HBM available etc. Otherwise it is hard to trust these numbers. Please take the time to segment the results into small vs large, dense vs MoE, relationship with context length etc.
2. Some recent literature reviews are missing. For example, [1]
3. The figures could be a bit better. For instance, Figure 5 is violating the margin.
4. The idea of the MIP is not used much throughout the paper, so casting the problem as a MIP seems a bit incomplete

**Questions:**

See weaknesses

---

> ### Author Response · Authors · 2025-11-18
> **Rebuttal**
>
> We thank the reviewer for their positive assessment and thoughtful comments.
>
> **Q1: The improvements claims made in the intro should be qualified by model type and size, context length, HBM available etc. Otherwise it is hard to trust these numbers. Please take the time to segment the results into small vs large, dense vs MoE, relationship with context length etc.**
>
> A: Thank you for the helpful suggestion. We have revised the claims in the introduction to clarify that the reported improvements are specific to particular model types and hardware configurations. In the paper, we now explicitly state the model type, context length, and GPU used for each reported number.
>
> **Q2: Some recent literature reviews are missing. For example, [1]**
>
> A: Thank you for pointing this out. We will be sure to include [1] in the final revision. However, we noticed that the citation details for [1] were not included in your comment. If possible, could you kindly provide the full reference? That would help us ensure the citation is accurate.
>
> **Q3: The figures could be a bit better. For instance, Figure 5 is violating the margin.**
>
> A: We appreciate your close attention to detail. We have resized it and a few others slightly to ensure clearer spacing and full compliance with the margin requirements. We appreciate again for your careful observation.
>
> **Q4: The idea of the MIP is not used much throughout the paper, so casting the problem as a MIP seems a bit incomplete**
>
> A: Thank you for the valuable comment. Our main motivation for initially introducing it as a MIP was that the routing decision variable $x$ is binary, and the objective (minimizing makespan), along with constraints such as Equations (1), (2), and (3) resemble a classic MIP structure. However, we agree that referring to it as a MIP may be indirect. Therefore, in the revised version, we have removed the mention of MIP at the beginning of Section 3.1 and now directly describe the problem as a makespan formulation, which we believe more accurately reflects the focus of our work.

---

### Author Response · Authors · 2025-11-26
**General Response**

Dear Reviewers and ACs,

We sincerely thank you for your positive assessments and valuable feedback throughout the review process! We have carefully addressed all the concerns raised in our rebuttal and revised the paper accordingly. The updated manuscript incorporates your constructive suggestions, with all changes marked in blue.

We believe the revisions have improved the quality and clarity of the paper. Please feel free to let us know if you have any remaining questions or concerns. We welcome further discussion!

Best regards,

The authors

---

### Author Response · Authors · 2025-12-02
**Summary**

Dear ACs and Reviewers,

Thank you for your time and effort throughout the review process, and for the consistently positive assessments and constructive feedback from all reviewers. Below, we summarize the core strengths of our work and how we addressed the main concerns raised in the reviews.

**Key strengths noted by reviewers**

- **Problem setup and Motivation**. Reviewers unanimously agreed that the problem of KV cache–aware load balancing is both **critical** and **timely**. They praised our **clear problem framing**, which explicitly couples cache eviction with routing and is **well-motivated from a practical LLM-serving perspective**. Reviewers also highlighted that our unified model **fills a critical gap** left by prior heuristic system designs.

- **Formal analysis and algorithm design.** Our main contributions are: (1) providing a **unified, theoretically grounded formulation** that explicitly couples cache eviction with global load balancing, bridging a critical gap left by prior heuristics and (2) offering a clear formalization with intuitive lemmas that lays the theoretical groundwork for our algorithms.

- **Evaluation Results.** Reviewers highlighted the **extensive and compelling** evaluation, which **covers diverse model architectures** (Dense and MoE, 8B-70B) and **workload scenarios**. They emphasized that the method demonstrates "**strong empirical results across the board**", consistently achieving high hit rates and throughput on diverse benchmarks.


**Key revisions addressing reviewer concerns**
- For Reviewer HKW6, the main concern is that the improvement claims in the introduction should be more specific and qualified. We have revised the corresponding parts to explicitly scope our performance improvements across different model types and sizes.

- For Reviewer NRb7, the primary requests focused on clarifying the practical representativeness of the assumptions in Theorem 5 and hyperparameter sensitivity. We fully addressed these by (1) justifying how these assumptions align with capacity limits and realistic batch diversity; and (2) demonstrating robustness by showing that all main experiments use a single fixed configuration across models and GPUs, and by adding an additional sensitivity study confirming performance stability. We also provided tuning-free guidelines for parameter selection.

- For Reviewer hRYq, the primary concerns are the stability of RLT under different random seeds and the scalability trends in Figure 6. We fully addressed these by (1) adding an ablation study showing that our method remains stable across a wide range of seeds and consistently outperforms the SOTA baseline, and (2) clarifying that, under a fixed request rate, all methods eventually saturate at the same maximum throughput equal to the request rate as the number of workers increases, while our method still achieves superior latency and hit rates.


- For Reviewer yivC, the primary request was to evaluate performance under different query-length distributions. We fully addressed this by (1) clarifying that our theory focuses on query-length heterogeneity in a worst-case sense and is agnostic to specific arrival distributions, (2) highlighting that the existing experiments already cover highly imbalanced query lengths, and (3) adding new experiments comparing uniform and heavy-tailed query-length distributions, which show that our method outperforms the strongest baseline in both cases and achieves even larger gains under heavy-tailed setting, thereby further confirming robustness to query-length distributions.


Best Regards,

The authors

---

### Meta-Review · Area_Chair_Aeg8 · 2026-01-11

**Summary:**

This paper demonstrates that previous LRU (Least Recently Used)-based KV cache management methods can be inefficient in online LLM serving scenarios. As a solution, the paper proposes a novel randomized KV cache eviction with learning-based routing methods. The experiments show significant advantages in latency and throughput compared to existing methods.

The AC and the reviewers agree that the paper has strong motivation, analysis, and experimental results.
Given the review and rebuttal, the AC recommends accepting this paper.

**Reviewer Concerns:**

- Reviewer HKW6 raised concerns about some unclear presentation of the paper. The rebuttal has addressed them accordingly.
- Reviewer NRb7 raised questions on the assumption of the theorem and hyper-parameter sensitivity. The rebuttal has addressed them accordingly.
- Reviewer hRYq requested additional experiments, and the rebuttal provides the results.
- Reviewer yivC also requested additional experiments, and the rebuttal provides the results.

Overall, there are no outstanding concerns at this time.

**Reviewer Scores:**

- Reviewer HKW6: 8->8
- Reviewer NRb7: 6->6
- Reviewer hRYq: 6->6
- Reviewer yivC: 6->6

---

### Decision · Program_Chairs · 2026-01-26

Accept (Poster)